# Meteorological and hydrological data from the Alder Creek watershed, SW Ontario

Andrew J. Wiebe[1,2], David L. Rudolph[1]

[1]Department of Earth and Environmental Sciences, University of Waterloo, Waterloo, N2L 3G1, Canada
[2]Department of Earth and Planetary Sciences, McGill University, Montreal, H3A 0E8, Canada

*Correspondence to*: Andrew J. Wiebe (ajwiebe@uwaterloo.ca)

**Abstract.** Data for small to mid-sized watersheds are seldom publicly available but may be representative of diverse types of hydrological contexts when assessing patterns. These types of data may also prove valuable for informing numerical experimentation and practical modelling. This paper presents data collected in the Alder Creek watershed, located within the
Grand River basin in Ontario, Canada. The Alder Creek watershed provides source water from the aquifers of the Waterloo Moraine for multiple well fields that supply the cities of Kitchener and Waterloo. Recharge rates and human impacts on streamflow are important topics for the watershed, and many numerical models of the area have been constructed. In order to support these types of analyses, field equipment was deployed within the watershed between 2013 and 2018 to monitor groundwater levels, stream stage, soil moisture, soil temperature, rainfall, and other weather parameters. The available data
are described, complementary information is presented, and examples of possible analyses are cited and illustrated. The data presented and described in this paper are available at https://doi.org/10.20383/101.0178 (Wiebe et al., 2019).

## 1 Introduction

Comprehensive meteorological and hydrological data from multiple field stations within small to mid-sized watersheds are seldom publicly available. This lack of data hinders the comparison of watersheds in different areas and the analysis of
hydrological patterns across the entire spectrum of watershed sizes. For instance, the spatial correlation structure of rainfall within a particular type of region may be poorly represented in the literature and therefore unavailable to verify or enhance regional models.

Moraines in southern Ontario are frequently used for public drinking water supply. Groundwater wells draw water for public supply from unconsolidated aquifers, which are replenished by (e.g., Lerner, et al., 1990; Wiebe, 2020): 1) rain and
snowmelt percolation through the vadose zone that arrives at the water table (diffuse recharge), 2) localized or depression focused recharge (DFR) that may occur in hummocky terrain, and 3) losing stream reaches (indirect recharge). This recharge is spatially variable and may vary in terms of its quality based on sources of contamination at the ground surface. The vulnerability of public supply wells to contamination is often assessed using numerical models, which require data ranging from groundwater levels to streamflow rates to precipitation amounts and evapotranspiration estimates. Groundwater recharge

is generally a calculated flux that occurs in the subsurface and is a major factor influencing simulated water levels; it is very difficult to measure directly and to quantify over large, heterogeneous areas.

The Alder Creek watershed represents many small watersheds where there are competing pressures related to groundwater. The watershed has multiple types of land use, including agriculture, aggregate (sand and gravel) extraction, and urban areas. These land use types each have their own groundwater quality and/or quantity concerns (Sousa et al., 2014).

Expanding urban development within the watershed is a major concern and a potential influence on groundwater recharge rates. Multiple public well fields are located within the watershed or capture water recharged within it (Brouwers, 2007), and these rely on maintaining groundwater recharge quantity and quality. There are ecological concerns regarding groundwater baseflow to the creek and how the public wells may influence this. Surficial geology data, stratigraphic data, and land use data are available for the watershed (see Section 8). Thus, the watershed is useful for assessing various critical issues related to

groundwater management due to the many important issues related to the watershed, and the amount of data available. This would include support for numerical modelling studies.

The Alder Creek watershed is a typical southern Ontario watershed and has been the subject of numerous studies (e.g., CH2MHILL and S.S. Papadopulos and Associates Inc., 2003; Matrix and S.S. Papadopulos and Associates Inc., 2014b; Sousa et al., 2013; Wiebe and Rudolph, 2020) due to its importance for local water supply. The Southern Ontario Water

Consortium (SOWC; now called the Ontario Water Consortium, www.ontariowater.ca) undertook to instrument the Alder Creek watershed as part of a project to set up a platform for testing new sensor technologies and to collect hydrological data. Research questions related to the installed equipment included topics such as how sensitive modelled groundwater recharge estimates might be to spatially variable rainfall (Wiebe and Rudolph, 2020), how recharge dynamics respond to extreme hydrological events (Menkveld, 2019), and how depression focused recharge might increase threats to public supply wells

(Wiebe, 2020; Wiebe et al., 2021). The Alder Creek watershed was the middle member of a continuum of three watersheds along a spectrum of urbanization that varied from fully urbanized (Mimico Creek, Greater Toronto Area, ON; 77 km$^2$; Toronto Region Conservation Authority, 2018) to rural/agricultural (Hopewell Creek, east of Kitchener, ON; 72 km$^2$; Irvine, 2018). Hydrological equipment was installed in the Alder Creek watershed between 2013 and 2018. The following summarizes the datasets that have been made available on the Federated Research Data Repository (www.frdr-dfdr.ca/repo/), shows example

graphs, and presents complementary information including borehole logs, piezometer construction descriptions, and analyses related to the dataset.

## 2 Site description

The Alder Creek watershed (Fig. 1; 78 km$^2$; e.g., described by Wiebe, 2020) is located on the Waterloo Moraine southwest of the cities of Waterloo and Kitchener in southern Ontario, Canada (43.3982° N, 80.5455° W). The Waterloo Moraine consists

of alternating coarse and fine layers of unconsolidated sediments (Martin and Frind, 1998) deposited at the confluence of multiple glacial ice lobe advances during the most recent deglaciation (Bajc et al., 2014). Alder Creek (a 4[th] order Strahler

stream; Ontario Ministry of Natural Resources and Forestry, 2015) is a tributary of the Nith River, which flows into the Grand River (basin area: 6,900 km$^2$). The quaternary geology of the watershed consists of sand and gravel units that are present over half of the watershed area, and less permeable units such as silt and clay glacial tills (Ontario Geological Survey, 2010).

Agriculture is the predominant land use (70 %) in the watershed (Ontario Ministry of Natural Resources, 2008; Region of Waterloo, 2010). The Alder Creek watershed is an important source of recharge and supplies source water for up to seven public well fields (Brouwers, 2007).

Figure 1 shows long−term and short−term monitoring locations near the watershed. A Water Survey of Canada (WSC) stream gauging station near New Dundee (02GA030; Water Survey of Canada, 2019) is located within the watershed on the

main branch of Alder Creek, and an Environment Canada weather station at Roseville (Government of Canada, 2019) reports temperature and precipitation < 3 km outside the watershed. The University of Waterloo weather station (weather.uwaterloo.ca; Seglenieks, 2020) also reports weather data in the area. Figure 2 provides the general context of major water budget components and shows that average monthly precipitation (Government of Canada, 2019) varies by up to 40 mm at the Roseville station and that the precipitation and potential evapotranspiration (PET) peaks occur in the same season, mid−

summer (Wiebe, 2020). Streamflow peaks in late winter around March. The baseflow index for the part of the watershed above the Water Survey of Canada gauge has been estimated to be 0.56 on average (Wiebe, 2020), and groundwater recharge has been estimated to be around 320 mm per year on average (M.H. Brouwers, pers. comm., 2017; Matrix and S.S. Papadopulos and Associates Inc., 2014a).


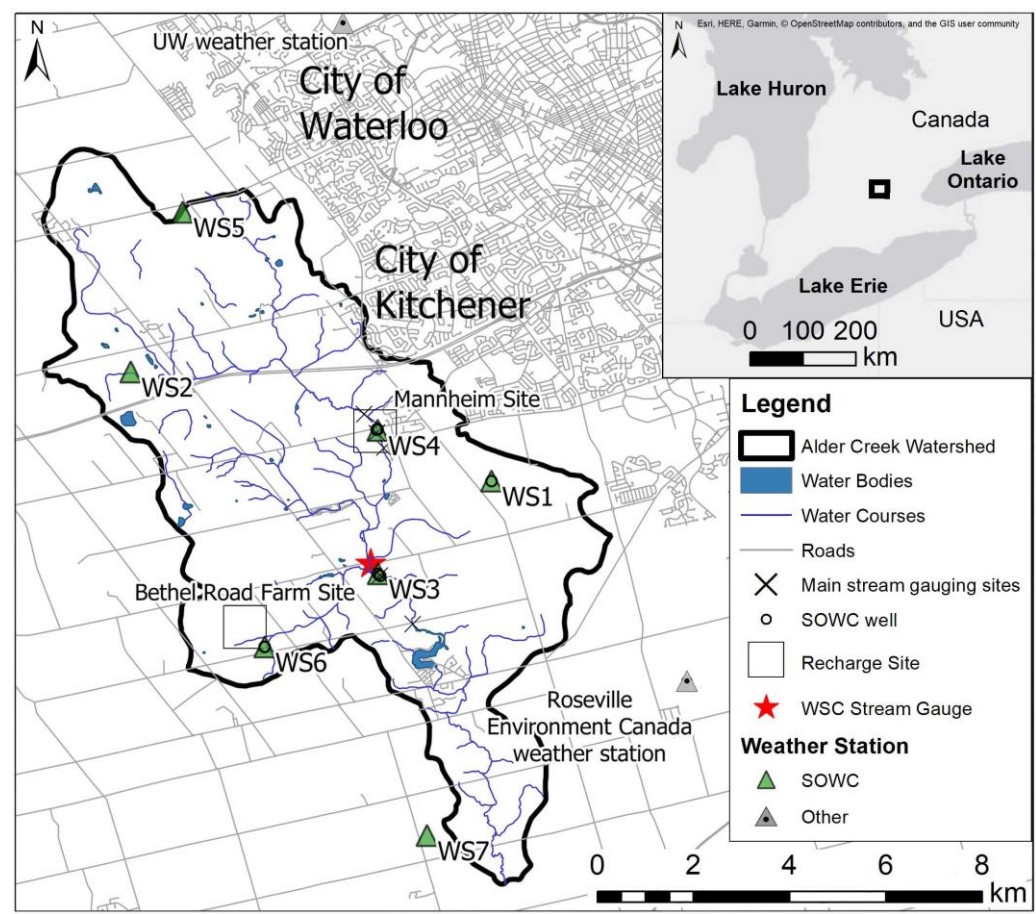

**Figure 1: Map of Alder Creek watershed and data collection locations (Esri et al., 2020b [Sources: Esri, HERE, Garmin, © OpenStreetMap contributors, and the GIS User Community]; DMTI, 2011; Government of Canada, 2019; Grand River Conservation Authority, 1998, Seglenieks, 2020; Wiebe et al., 2019). Seven weather stations and two recharge stations were installed to measure meteorological and hydrological data. Complementary datasets are available from the Water Survey of Canada (WSC)**
**stream gauge (Water Survey of Canada, 2019), the Roseville Environment Canada weather station (Government of Canada, 2019), and the University of Waterloo (UW) weather station (Seglenieks, 2020), in addition to other sources, as noted in Section 8.**

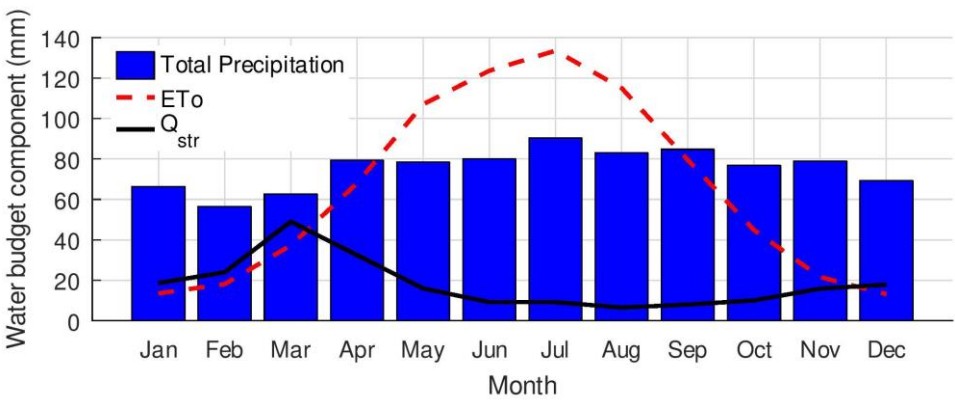

**Figure 2: Major water budget component estimates for the Alder Creek watershed (adapted from Wiebe, 2020; based on data from: Government of Canada, 2019; Water Survey of Canada, 2019; and Wiebe et al., 2019). Reference ET (ETo) estimates were calculated via the Penman−Monteith method (Raes, 2009). Streamflow ($Q_{str}$) was estimated for the watershed outlet based on a scaling factor. Groundwater recharge has been estimated to be around 320 mm per year (M.H. Brouwers, pers. comm., 2017; Matrix and S.S. Papadopulos and Associates Inc., 2014a), and the baseflow index has been estimated to be 0.56 for the Water Survey of Canada gauge (Wiebe , 2020).**

Equipment installations during the Southern Ontario Water Consortium – Alder Creek project included weather stations, recharge stations, and stream stations (Fig. 1). Seven weather stations were installed in and around the watershed. Two sites (the Mannheim site and the Bethel Road Farm site; Fig. 1) were instrumented to monitor vadose zone drainage and groundwater levels, and to estimate recharge rates. These recharge sites are shown in more detail in Fig 3. Twenty observation wells (including three drive−point streambed piezometers) were installed at the Mannheim site, and 15 observation wells were installed at the Bethel Road Farm site. Soil moisture and subsurface temperature were monitored at these two sites. Five locations along the creek were instrumented with pressure transducers to monitor surface water levels and temperatures. Stream gauging and the development of rating curves were conducted for these locations to augment the records at the Water Survey of Canada gauge within the watershed.

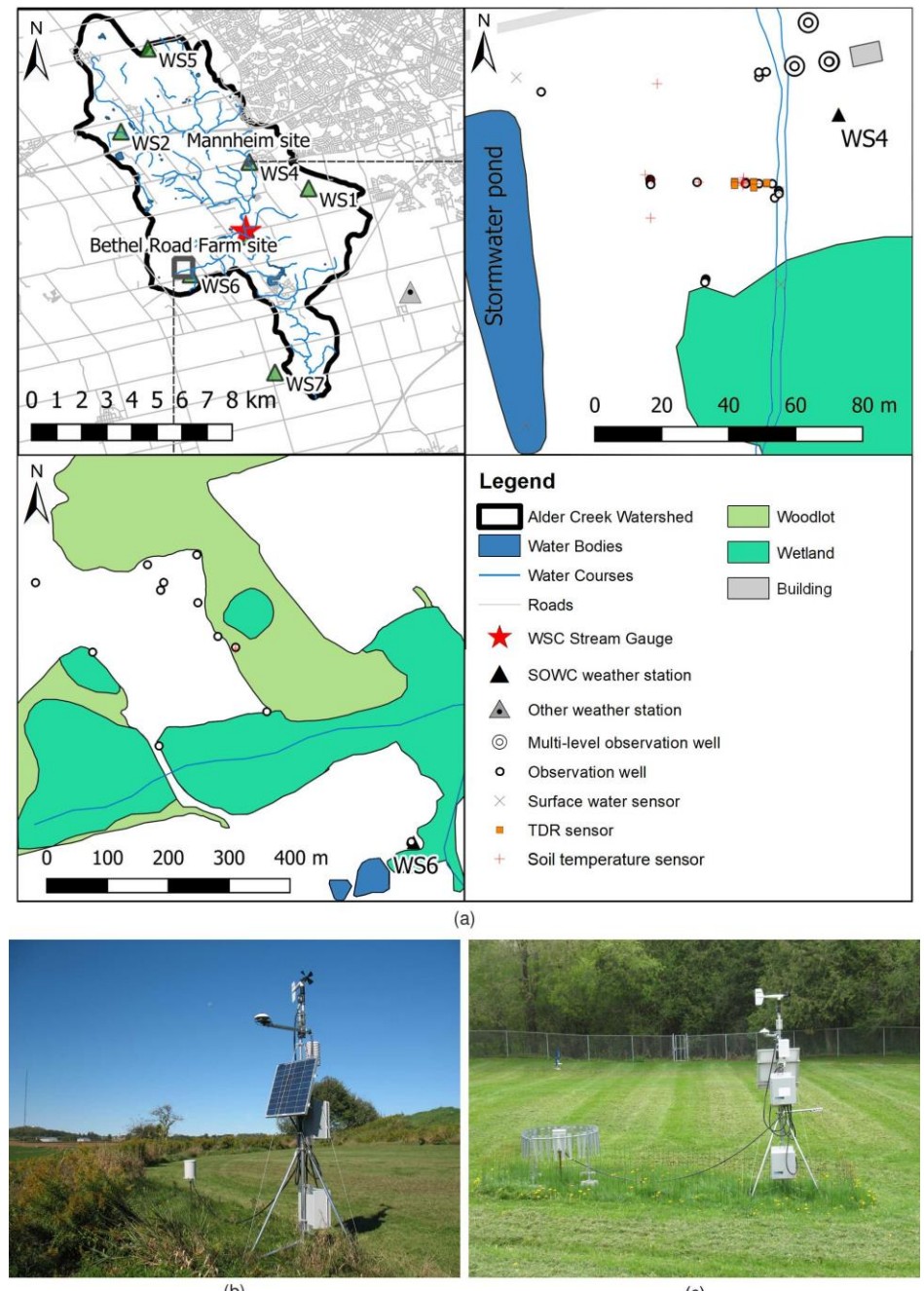

**Figure 3: a) The Mannheim and Bethel Road Farm "Recharge" sites (woodlots, wetlands, and buildings adapted from Esri et al., 2020a [Sources: Esri, HERE, Garmin, Intermap, INCREMENT P, GEBCO, USGS, FAO, NPS, NRCAN, GeoBase, IGN, Kadaster NL, Ordnance Survey, Esri Japan, METI, Esri China (Hong Kong), © OpenStreetMap contributors, GIS User Community]; DMTI, 2011; Government of Canada, 2019; Grand River Conservation Authority, 1998; Wiebe et al., 2019), and photos of stations b) WS2, and c) WS4. Station WS2 is shown prior to installation of windscreen around rain gauge. Groundwater levels, groundwater temperatures, soil temperatures, and soil moisture were monitored at both sites (moisture was monitored concurrently with temperature at the Bethel Road Farm site).**

## 3 Meteorological data

Seven weather stations (Fig. 1; WS1 to WS7) were installed during the Southern Ontario Water Consortium – Alder Creek project to monitor spatially variable precipitation and parameters related to evapotranspiration in the area. Figure 3 shows photos of two weather stations as examples, and the components of the stations are listed in Table 1. Data were typically downloaded hourly from the stations to a computer at the University of Waterloo (Waterloo, Ontario, Canada) via the cellular telephone network.

Data from the weather stations was reviewed and missing time stamps were assigned placeholders (e.g., "#N/A") to indicate "not available." Anomalous data values, i.e., values outside of an acceptable range for the parameter and the season, were similarly assigned placeholders. Despite this, occasional erroneous values and error codes may still be present, and the data should be reviewed for quality prior to use.

Figure 4 presents example weather data and derived (Penman−Monteith) PET (Raes, 2009) estimates at WS2. Similar
records are available for each of the seven stations, although there are systemic differences depending on where the stations were located within the watershed. Average air temperature (e.g., Fig. 4a) derived from 15 min time intervals ranged from -34.7 °C to +32.9 °C between Jan 2014 and Dec 2018, based on data from all seven stations. Consistent differences between temperatures at the seven weather stations could be related to the positioning of each sensor relative to local vegetation. Relative humidity (e.g., Fig. 4b) ranged from 16 to 100 % across all stations. Wind speeds (Fig. 4c) at the seven weather
stations varied from 0 m·s$^{-1}$ to a maximum of 14.6 m·s$^{-1}$ (WS3), with an overall average of 1.6 m·s$^{-1}$. Knowledge of the average wind speed was helpful during calculations of PET (Wiebe, 2020) when specifying a value to fill−in for missing information. Solar radiation (Fig. 4d) was typically measured as incoming solar radiation with solar pyranometer devices and ranged up to 1,190 W·m$^{-2}$ at the seven weather stations. Net incoming radiation estimates at WS7 were up to 893 W·m$^{-2}$. Figure 4e shows an example of the daily Penman−Monteith PET estimates derived from the WS2 data using the ETo Calculator program (ETo:
reference evapotranspiration; Raes, 2009).

**Table 1: Meteorological variables and instruments in the Southern Ontario Water Consortium − Alder Creek network.**

| Station | WS1 | WS2/WS3/WS4/WS5 | WS6 | WS7 |
|---|---|---|---|---|
| Record[*] | Jan 2014 – Dec 2016 | | | |
| Air temperature (°C) and relative humidity (%) | HC2-S3 | HMP155 | HMP45C / HMP155[§] | HMP45C |
| Wind speed (m·s⁻¹) and wind direction (° from North) | R.M. Young 05103 | R.M. Young 05103 | R.M. Young 05103 | R.M. Young 05103 |
| [Measurement height above ground] | [3 m] | [3 m] | [3 m] | [3 m] |
| Rainfall (rain gauge) | TE525WS / TB3[†] | TB3 | TB3[‖] (×2) | TE525WS / TB3[‖] |
| [Measurement height above ground] | [1 m] | [1 m] | [1 m] | [1 m] |
| Snowfall (sonic snow depth) | SR50A (Nov 2014 to Apr 2015 only) | SR50A (Nov 2014 to Apr 2015 only) | SR50A (Nov 2014 to Apr 2015 only) | SR50A (Nov 2014 to Apr 2015 only) |
| Incoming solar radiation (W·m⁻²) | SPLite2 | EQ08-SE | SPLite2/ EQ08-SE[§] | CNR1[¶] |
| Barometric pressure | N/A[‡] | CS106 (WS4 only) | N/A | Young 61205V |
| Datalogger | CR1000 | Sutron 9210B | CR1000/ Sutron 9210B[§] | CR23X / CR1000 |
| Additional instruments | Observation well: water level and temperature – PT12® | N/A | Observation wells: water levels and temperatures – PT12® | Soil moisture – CS616 |
| Telemetry | RAVEN X-HSPA cellular modem | BT6801 cellular modem | BT6801 cellular modem | BT6801 cellular modem |

[*] Highest quality period of record

[†] Rain gauge switched to the latter on 6 May 2014

[‡] N/A – not applicable

[§] Datalogger switched to the latter instrument on 5 Jun 2014

[‖] Rain gauge switched to the latter on 11 Jul 2016

[¶] WS7 used a Campbell Scientific CNR1 Net Radiometer to measure incoming and outgoing short wave and long wave solar radiation (via pyranometer and pyrgeometer)

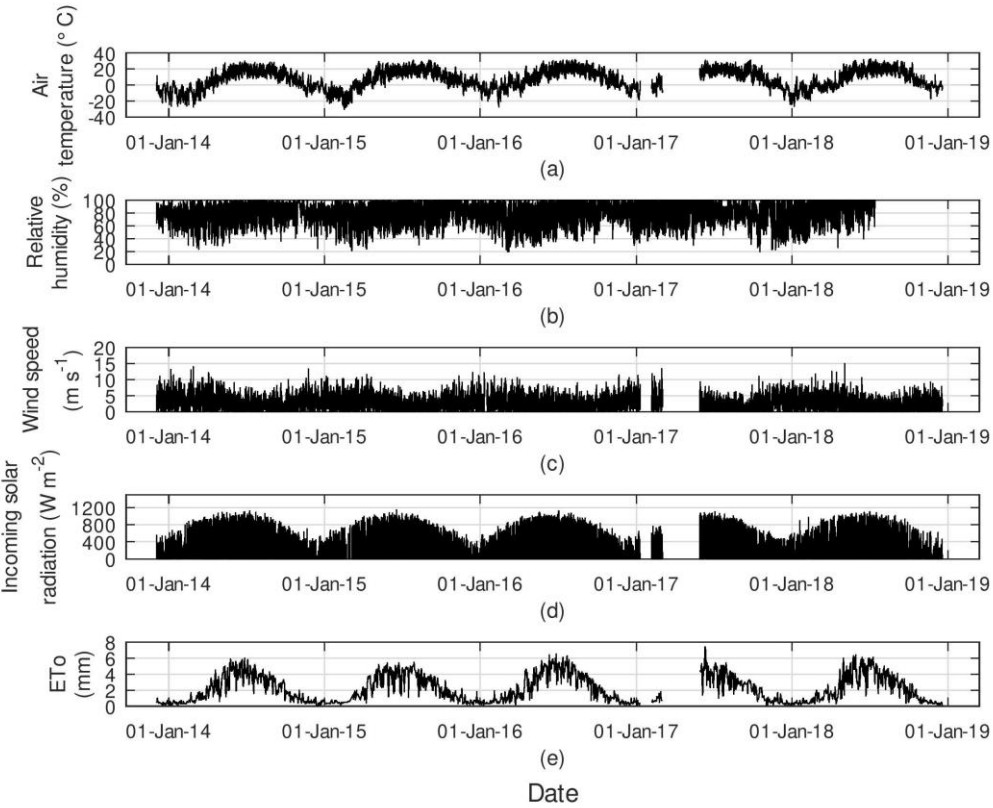


**Figure 4: WS2 data (Wiebe et al., 2019) and derived Penman−Monteith PET (Raes, 2009) estimates: (a) air temperature, (b) relative humidity, (c) wind speed, (d) solar radiation, and (e) reference evapotranspiration.**

Rainfall (e.g., Fig. 5) was monitored at each of the weather stations using a tipping bucket rain gauge (either Texas Instruments TE525 or Hydrological Services TB3). Each gauge was installed at a height of 1 m above ground surface and surrounded with a 1 m radius, Alter−type wind screen. A second gauge was additionally installed at station WS6 with two concentric wind screens at radii of 1 m and 2 m. Faulty wiring prevented reasonable rainfall data from being recorded at station WS1. Spatial correlation quantified via Spearman Rank Correlation coefficients among the six other weather stations and the Roseville Environment Canada weather station was between 0.5 and 0.8 (Wiebe, 2020).


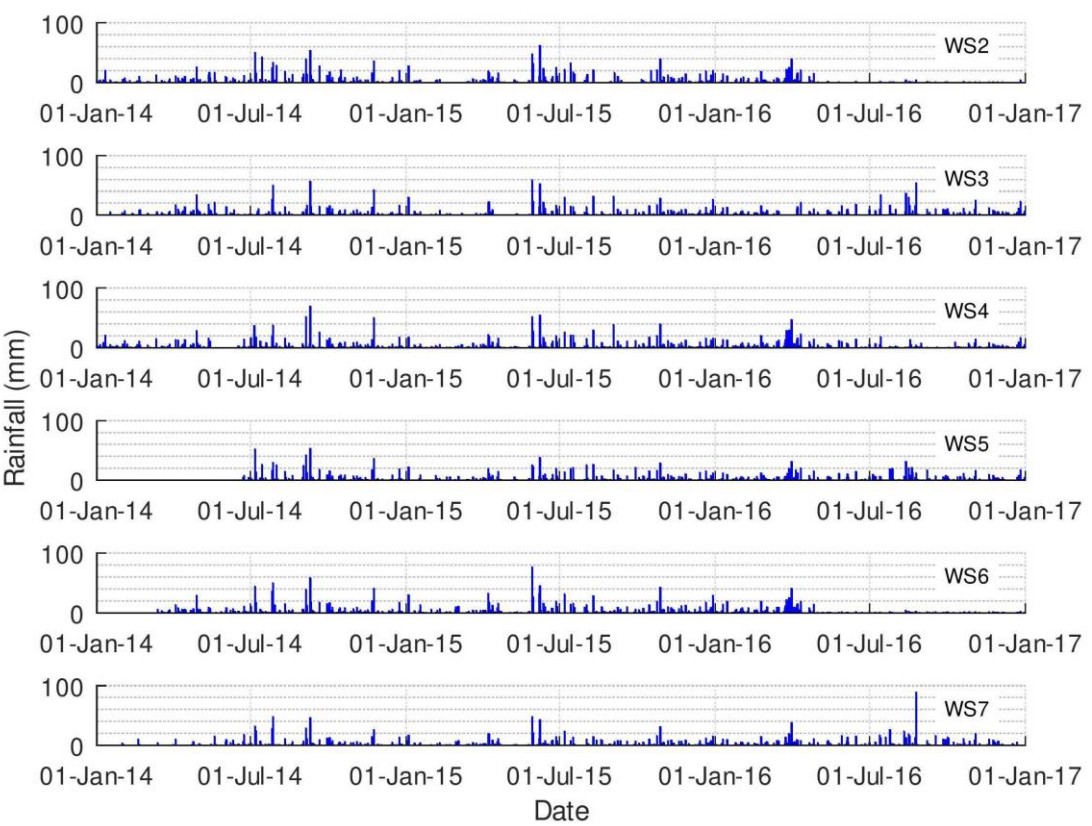

**Figure 5: Daily rainfall at stations WS2 to WS7 (Wiebe et al., 2019). Annual rainfall ranged from around 400 to 1000 mm at these six stations.**

Snowfall was monitored via sonic sensors that estimated snow depth above ground surface. The sonic snow sensors at stations WS2 to WS7 reported data for the 2014 to 2015 winter season. The snow depth data collected agree well with observations at the Roseville Environment Canada station (Fig. 6). The average from these six stations was within 2 cm of the Roseville amount on an event−by−event basis (Wiebe, 2020). Due to maintenance issues, the sonic snow sensors were only deployed for the one winter season. Because declines in snowpack thickness may indicate increases in snowpack density rather

than the release of meltwater (Dingman, 2015), estimating the timing of snowmelt is often desirable. Wiebe et al. (2021) used a degree-day method (Rango and Martinec, 1995) to estimate snowmelt for the Alder Creek watershed, and the calculations are described in the supplementary materials associated with that paper.

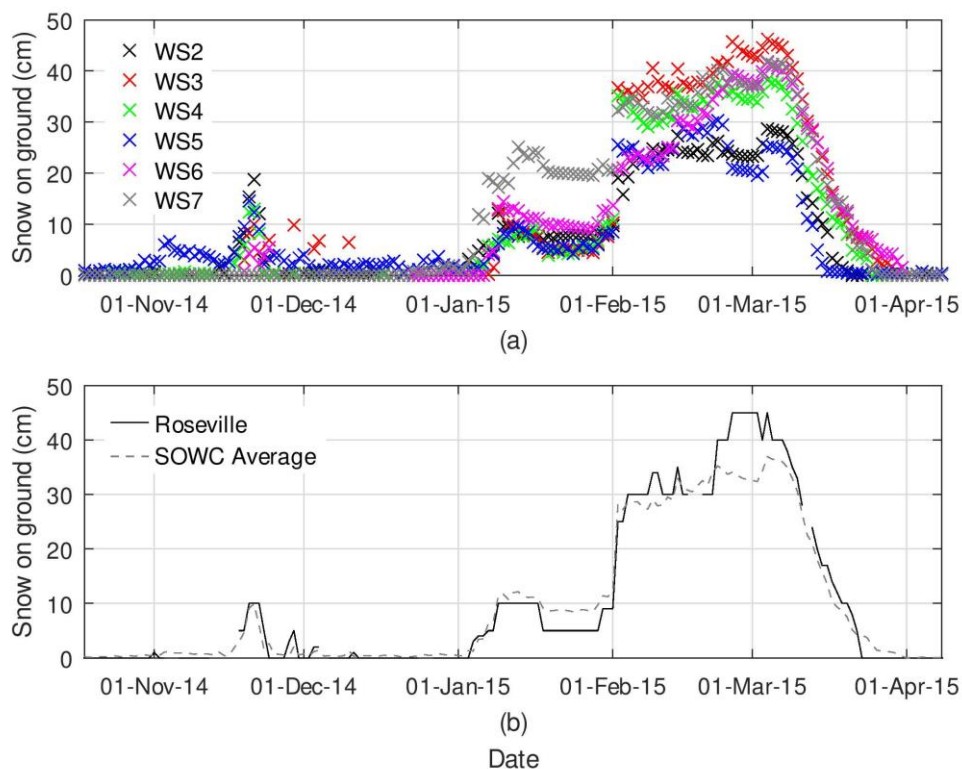

Figure 6: Snowpack thickness data (a) for individual Southern Ontario Water Consortium (SOWC) stations WS2 – WS7 (Wiebe et al., 2019), and (b) for the average from the SOWC stations and for Roseville (Government of Canada, 2019) for the 2014 to 2015 winter season (adapted from Wiebe, 2020). The average snowpack thicknesses are generally close to the Roseville weather station records, suggesting that uniform snowfall may be a reasonable assumption for the Alder Creek watershed.

## 4 Groundwater data

Multi−level or single−screen observation wells were installed at several locations within the watershed. Borehole logs are discussed in the next section. Pressure transducers – either vented (AquiStar PT12®) or non−vented (Solinst Levelogger®) – were installed in most of the observation wells. The multi−level Solinst "Continuous Multichannel Tubing" (CMT®) wells did not have pressure transducers. Manual water level measurements were made occasionally at the wells to track water levels in the wells without pressure transducers, or to provide adjustment targets for the pressure transducer data. Figure 7 shows an example of PT12® (AquiStar Inc.) pressure transducer data from CPP3 at the Mannheim site that were adjusted based on the average offset from manually measured water level elevations, where the sensor measurement point was originally assumed equivalent to the bottom elevation of the piezometer screen. Solinst Levelogger® data would require the additional

intermediate step of barometric pressure compensation (Solinst, 2020). Water levels fluctuated over an amplitude range (maximum minus minimum water level) of up to 1.2 m over an annual cycle at background locations at the Mannheim site, i.e., locations not expected to be affected by DFR or indirect recharge beneath the stream (Wiebe, 2020). Water levels at observation wells affected by DFR (e.g., CPP2) fluctuated over a range of up to 2.4 m (Wiebe, 2020).

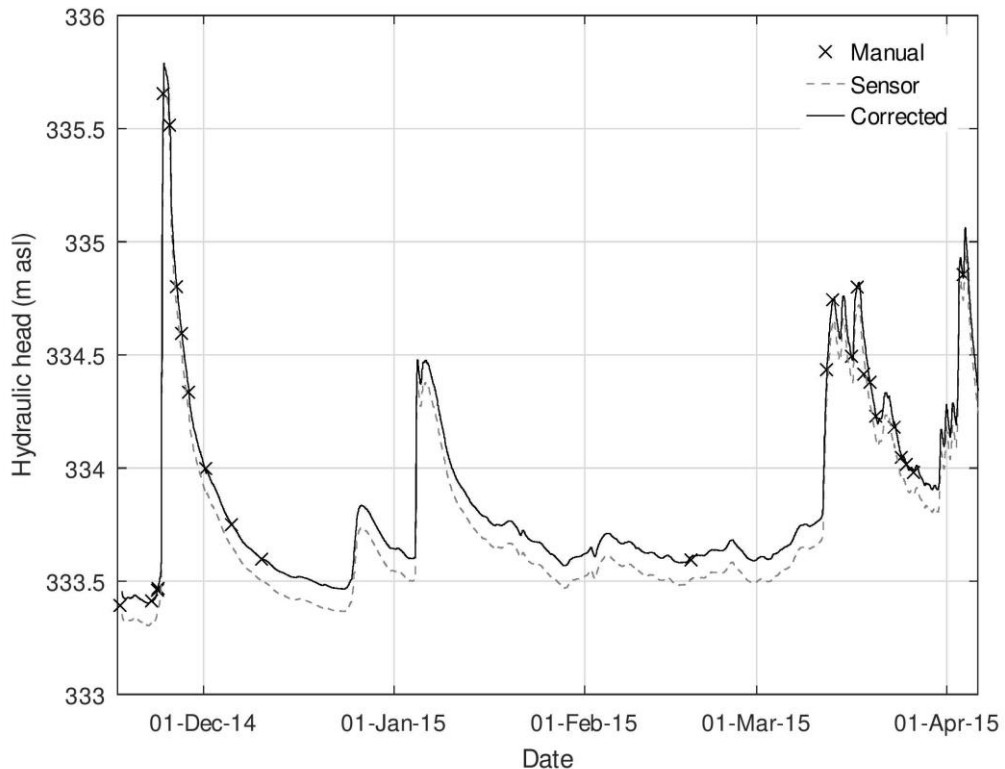

**Figure 7: Adjustment of CPP3 (Mannheim site) pressure transducer data based on manual levels (Wiebe et al., 2019), with spikes due to sampling removed. Note: "m asl" indicates** *metres above sea level***.**

Groundwater temperature data were recorded coincident with water levels at most of the larger−diameter wells. Alder Creek appears to be a losing stream reach during at least part of the year at the Huron Road Farm site and at the Mannheim site, based on the manual CMT water level data. Readings in the streambed drive−point piezometers at the Mannheim site occasionally indicated the presence of unsaturated soil between surface water and the water table (i.e., a water tape reading could not be made because the observation well was dry).

**5 Vadose zone data**

Sediment samples were collected, and borehole logs were drawn for some locations. A limited number of grain size analyses were also conducted. Soil moisture and soil temperature data were collected at the two Recharge sites within the Alder Creek watershed. Two different temperature methods were employed to illustrate how the data may be used to estimate vadose zone drainage rates at the Mannheim site, as discussed below.

**5.1 Soil texture**

The availability of borehole logs is summarized in Table 2. Borehole logs for the observation wells installed in the Alder Creek watershed were not included with the dataset, but ten logs are included below and several are available elsewhere, as indicated in Table 2. Figure 8 shows the logs for three deeper wells that are described in Table 3, and Fig. 9 shows seven shallower logs from the Bethel Road site that are described in Table 4. Grain size analyses were conducted for soil samples from cores at

three locations. Figure 10 shows results from selected coarser depth intervals at one location at the Mannheim site and at two locations at the Bethel Road Farm site. The borehole log for MLT1 (Wiebe, 2020) at the Mannheim site suggests silt or silty sand present in most of the borehole, with two coarser, sandy sections around depths of 0.4 m and 3.1 m. The grain size analyses for this borehole support the interpretation of poorly graded, gravelly sand present around a depth of 0.4 m, and the interpretation of silty sand throughout other sections. The grain size curves for Bethel Road Farm MLT1 correspond to the

associated borehole log (see Fig. 9), with finer material (sandy silt) in the uppermost sample, gravelly sand at intermediate depths, then a more homogeneous medium sand. The grain size curves for the core adjacent to P3 at Bethel Road Farm mostly represent fine to medium sand and generally correspond to the borehole log (Fig. 9). The gravelly sand layer from 3.05 m to 4.60 m depth is well represented by the curve for 3.30 m to 3.40 m, though the subsequent lower interval represents a more homogeneous fine sand lens or unit present within the layer.


**Table 2: Overview of the availability of borehole logs related to the Southern Ontario Water Consortium – Alder Creek project.**

| Site / weather station | Observation well/instrument name (Provincial well tag #) | Reference |
|---|---|---|
| Mannheim | CMT1, CMT2a, CMT2b, CMT3 | Hillier (2014) |
| Mannheim | Boreholes near CPP1, CPP2, CPP6, and CPP8; and at MLT1 | Menkveld (2019); Appendix G in Wiebe (2020) |
| Bethel Road | M1, MLT1, P3, P5, P6, P7, P13 | This article |
| WS1 | (A151035) | Province of Ontario (2021), this article |
| WS3 | CMT4 (A155063), CMT5 (A155050) | Province of Ontario (2021), this article |
| WS6 | (A155083), (A155084) | Province of Ontario (2021), this article |

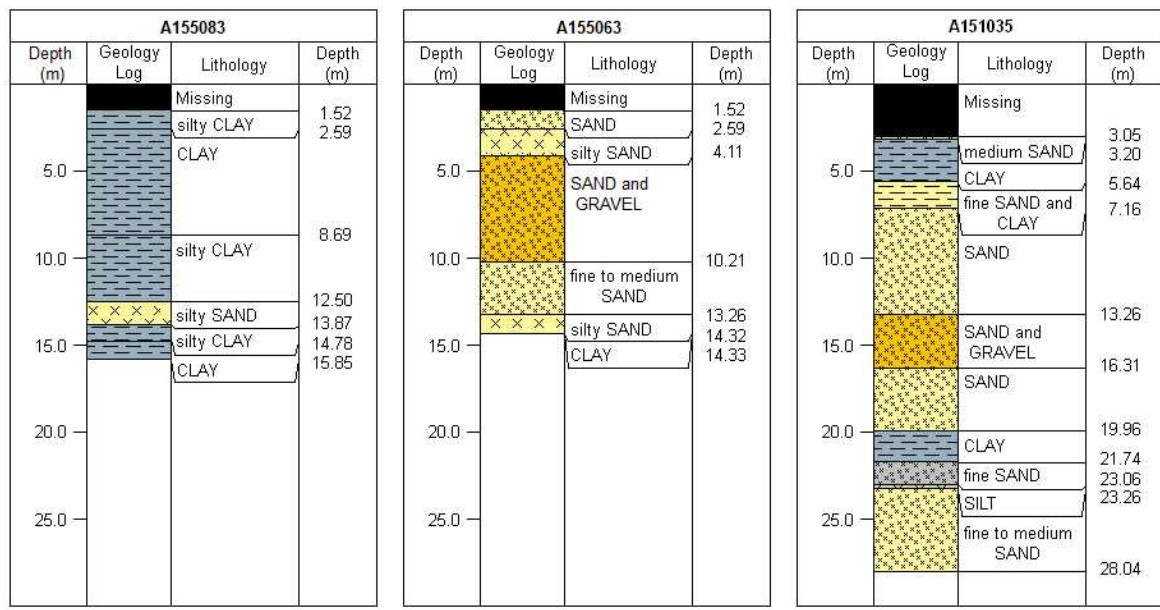

BGS Groundhog® Desktop, Copyright © BGS/UKRI (2021)

**Figure 8: Deeper (> 10 m) borehole logs (Natural Environment Research Council, 2017). Splitspoon samples (length: 61 cm) were collected at the top of each 1.52 m interval and then interpolated and summarized to produce these logs.**

**Table 3: Deeper (> 10 m) borehole logs and monitoring well details. This table provides complementary information for the Wiebe et al. (2019) dataset.**

| Site / weather station | Trussler Road Farm / WS1 | Huron Road Farm / WS3 | Bethel Road Farm / WS6 |
|---|---|---|---|
| Provincial well tag # | A151035 | A155063[‡] | A155083[¶] |
| Easting (m)* | 538896.06 | 536584.58 | 534185.28 |
| Northing (m)* | 4803972.07 | 4802004.55 | 4800526.67 |
| Ground surface elevation (m asl)* | 371.02 | 321.88 | 339.88 |
| Top of casing elevation (m asl)* | 372.15 | 323.13 | 340.72 |
| Drilling start date | 06 Feb 14 | 12 Feb 14 | 14 Feb 14 |
| Drilling completion date | 10 Feb 14 | 13 Feb 14 | 18 Feb 14 |
| Depth of borehole (m bgs[†]) | 28.04 | 14.33 | 15.85 |
| Screened interval (m bgs[†]) | 18.29−19.81 | 0.76−0.87, 2.74−2.85, 4.75−4.86, 6.71−6.82, 8.69−8.80, 10.64−10.75, 12.73−12.84 | 7.62−9.14 |
| Soil sampling | | Split spoon (length: 0.61 m), one sample approximately every 1.5 m | |
| Type of well | 0.051 m diameter PVC | 7−port Solinst CMT® | 0.051 m diameter PVC |
| Backfill materials within borehole annulus space around casing | bentonite chips (17.7−0.6 m bgs), then cement up to surface | Sand (water table – 1.5 m bgs), then bentonite chips up to ground surface | Sand (9.14−7.62 m bgs), then bentonite chips up to ground surface |

* Datum: NAD83, Zone 17N; "m asl" indicates "metres above sea level"

[†] "bgs" indicates "below ground surface"

[‡] Similar installation for well A155050, except that: the well top of casing coordinates were (536540.35 m E,4802045.6 m N, 323.24 m asl); the ground surface elevation was 322.48 m asl; the borehole depth was 15.02 m bgs; and the screened intervals were 2.80−2.91 m bgs, 4.75−4.86 m bgs, 6.66−6.77 m bgs, 8.75−8.86 m bgs, 10.72−10.83 m bgs, 12. 69−12.80 m bgs, and 14.65−14.76 m bgs.

[¶] Similar installation for well A155084, except that: the well coordinates were (534185.94 m E, 4800529.31 m N, 340.84 m asl); the ground surface elevation was 340.09 m asl; the borehole depth was 4.57 m; the screened interval was 3.05 m bgs to 4.57 m bgs; sand was backfilled from 4.57 m bgs to 3.05 m bgs, and bentonite was backfilled from 3.05 m bgs up to ground surface.

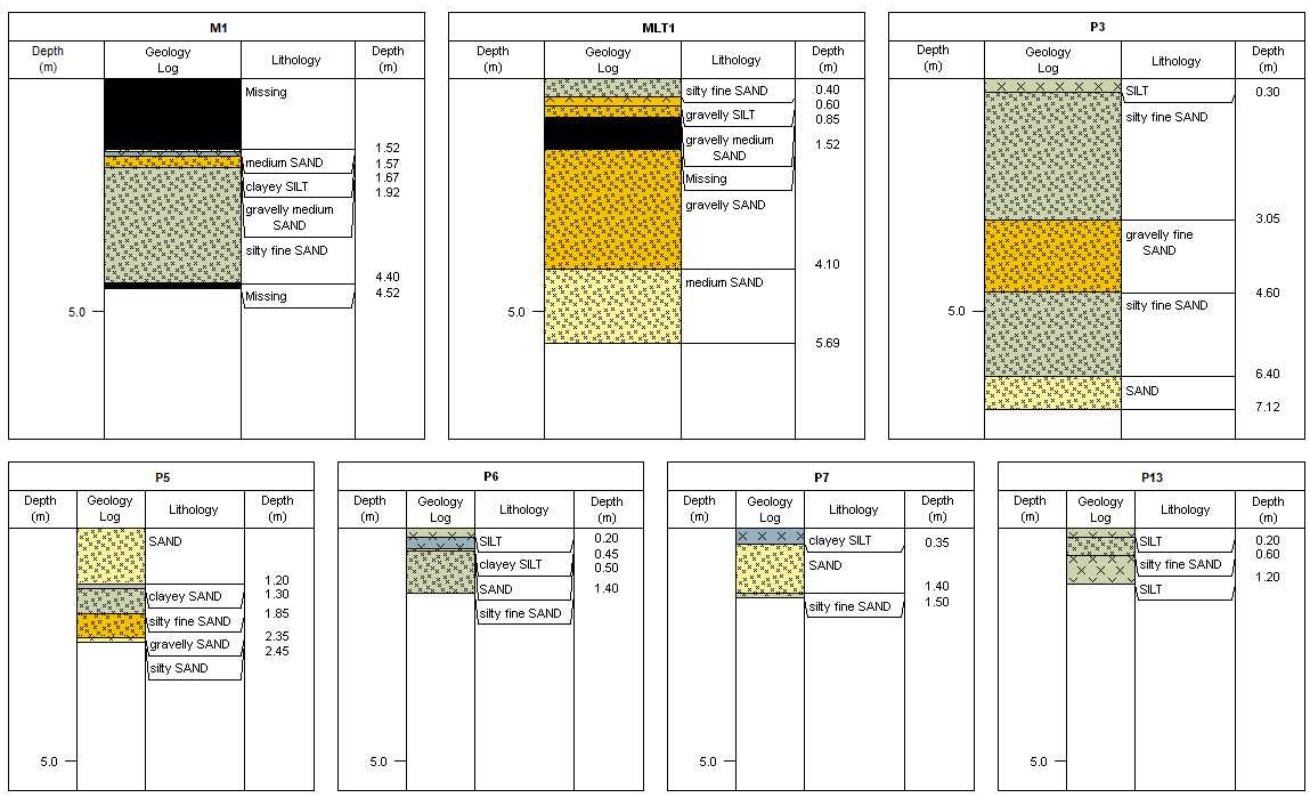

**Figure 9: Shallow (< 10 m) borehole logs (NERC, 2017). The borehole logs for M1, MLT1, and P3 were based on the analysis of cores (length: 1.52 m) from a continuous coring method (7720DT GeoProbe®), while the borehole logs for P5, P6, P7, and P13 were based on analysis of a continuous series of samples (length: about 15 cm) collected with a hand auger.**

**Table 4: Bethel Road Farm shallow piezometers (P1 to P13) and borehole logs. This table provides complementary information for the Wiebe et al. (2019) dataset.**

| Name | Bottom Depth (m bgs) | Screen Length (m) | Extension of casing above ground surface (m) | Diameter (m) | Installation Date | Comments |
|------|------|------|------|------|------|------|
| P1 | 4.61 | 0.3 | 0.263 | 0.0254 | 08 Dec 15 | Under active agricultural field |
| P2 | 5.60 | 0.3 | 0.8 | 0.0254 | 08 Dec 15 | Under active agricultural field |
| P3 | 7.92 | 0.3 | 0.87 | 0.0254 | 08 Dec 15 | Woodlot recharge plot; borehole log collected from core 1 m away |
| P4 | 7.01 | 0.3 | 0.97 | 0.0254 | 08 Dec 15 | Woodlot recharge plot |
| P5 | 2.67 | 0.3 | 0.7 | 0.0254 | 08 Dec 15 | |
| P6 | 1.43 | 0.3 | 0.43 | 0.0254 | 09 Dec 15 | |
| P7 | 1.32 | 0.3 | 0.54 | 0.0254 | 08 Dec 15 | |
| P8 | 12.77 | 0.3 | 0.954 | 0.0254 | 09 Dec 15 | Screened in medium sand |
| P9 | 7.42 | 0.3 | 0.45 | 0.0254 | 09 Dec 15 | |
| P10 | 4.46 | 0.3 | 0.41 | 0.0254 | 09 Dec 15 | |
| P11 | 5.10 | 0.3 | 1.28 | 0.0254 | 09 Dec 15 | |
| P12 | 3.23 | 0.3 | 0.43 | 0.0254 | 09 Dec 15 | |
| P13 | 1.09 | 0.3 | 0.7 | 0.0254 | 16 Dec 15 | |
| M1[*] | 4.52 | - | - | - | 08 Dec 15 | Borehole log; water table encountered around 1.3 m bgs[†] |
| MLT1 | 5.69 | - | - | - | 16 Dec 15 | Borehole log; water table encountered around 4.1 m; multi−level tensiometer device installed – see Appendix C in Wiebe (2020) |

[*] Approximate coordinates: (533820 m E, 4800830 m N, 343 m asl). Coordinates for the other instruments are listed in the Wiebe et al. (2019) dataset.

[†] "bgs" indicates "below ground surface"

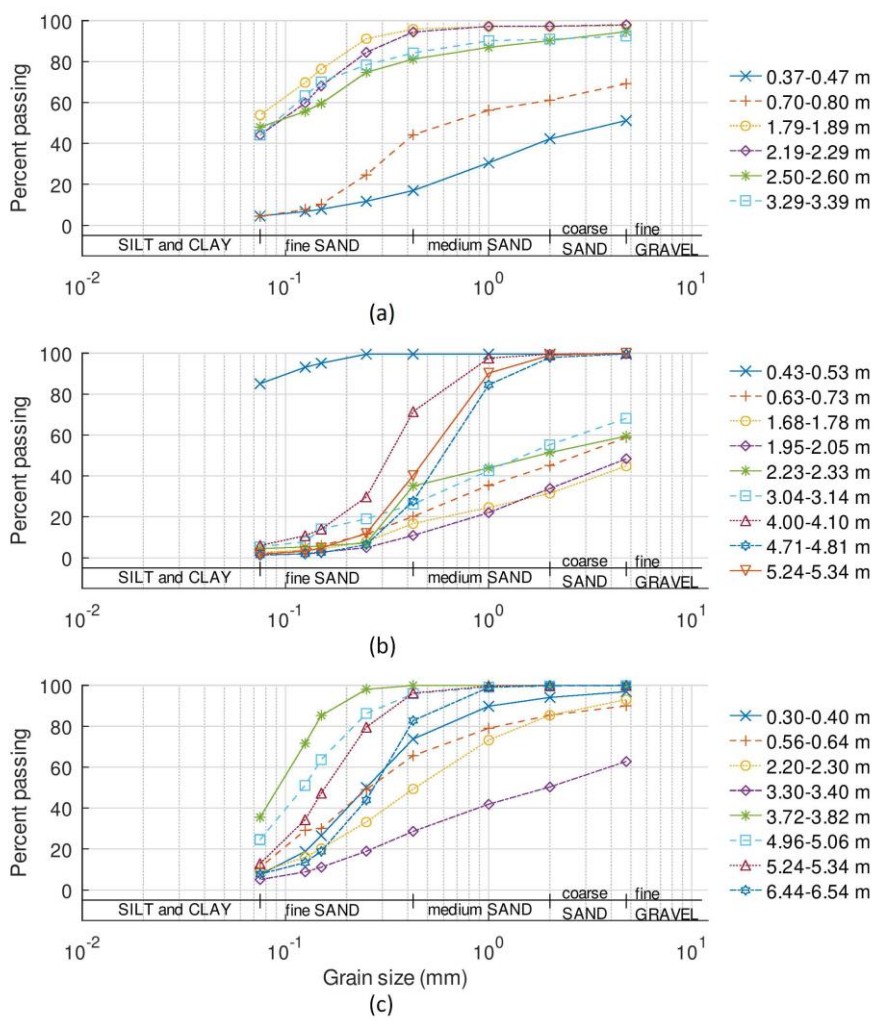

**Figure 10: Grain size analyses at (a) Mannheim MLT1, (b) Bethel Road Farm MLT1, and (c) the soil core within 1 m of Bethel Road Farm P3. Soil sample depth intervals are listed in metres below ground surface. The ground surface elevations for these locations are approximately 336.15 m asl, 350.61 m asl, and 349.70 m asl, respectively.**

260

## 5.2 Soil moisture

Soil moisture (e.g., Fig. 11) was monitored via two sets of eight instruments at the Mannheim site, and one set of eight instruments at the Bethel Road Farm site. Time domain reflectometry (TDR; 0.3 cm sensor length; TDR100, Campbell Scientific Inc.) sensors and water content reflectometer (0.12 m sensor length; CS655, Campbell Scientific Inc.) instruments
265 were installed at depths between 0 m and 1.5 m at the Mannheim site, and the water content reflectometer sensors were also installed at depths between 0 and 1.11 m at the Bethel Road Farm site (Table 5).

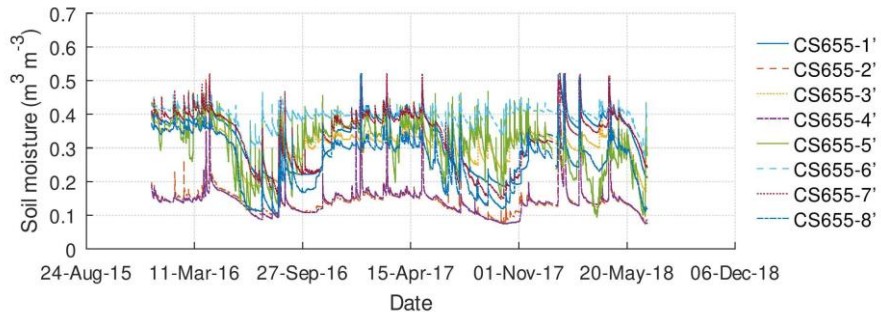

**Figure 11: Example of soil moisture data from the Mannheim site (sensor depths: between 0.15 and 0.63 m below ground surface).**

270

**Table 5: Soil moisture sensors.**

| Site | Sensor name | Angle from vertical (°) | Total distance along angle to end of probe (m) | Vertical depth of top of sensor (m) | Vertical depth of bottom of sensor (m) |
|---|---|---|---|---|---|
| Mannheim | TDR1 | 0 | - | 0.00 | 0.30 |
| | TDR2 | 0 | - | 0.31 | 0.61 |
| | TDR3 | 0 | - | 0.00 | 0.30 |
| | TDR4 | 0 | - | 0.30 | 0.60 |
| | TDR5 | 0 | - | 1.20 | 1.50 |
| | TDR6 | 0 | - | 0.00 | 0.30 |
| | TDR7 | 0 | - | 0.30 | 0.60 |
| | TDR8 | 0 | - | 0.61 | 0.91 |
| Mannheim | CS655-1' | 45 | 0.5 | 0.27 | 0.35 |
| | CS655-2' | 45 | 0.69 | 0.40 | 0.49 |
| | CS655-3' | 45 | 0.44 | 0.23 | 0.31 |
| | CS655-4' | 45 | 0.89 | 0.54 | 0.63 |
| | CS655-5' | 0 | - | 0.00 | 0.12 |
| | CS655-6' | 0 | - | 0.15 | 0.27 |
| | CS655-7' | 0 | - | 0.38 | 0.50 |
| | CS655-8' | 0 | - | 0.19 | 0.31 |
| Bethel Road Farm | CS655-1 | 45 | 0.68 | 0.40 | 0.48 |
| | CS655-2 | 45 | 1.11 | 0.70 | 0.78 |
| | CS655-3 | 45 | 1.52 | 0.99 | 1.07 |
| | CS655-4 | 0 | - | 0.37 | 0.49 |
| | CS655-5 | 0 | - | 0.66 | 0.78 |
| | CS655-6 | 0 | - | 0.99 | 1.11 |
| | CS655-7 | 0 | - | 0.69 | 0.81 |
| | CS655-8 | 0 | - | 0.84 | 0.96 |

**5.3 Soil temperature**

Vadose zone temperature profiles were monitored via three approaches. In the first approach, six or seven Tidbit v2 (Onset Computer Corp.) temperature sensors were fixed onto each of three 2.54 cm diameter solid stem PVC rods at intervals and then the rods were installed into separate boreholes drilled using a 7720DT GeoProbe® drill rig with a direct−push system. The three boreholes were installed in locations where different conditions were expected at the Mannheim site (e.g., beneath anticipated ponding in the base of the topographic depression, and at background locations with higher elevations). The sensors on each rod recorded temperatures at depths between 0.1 m and 1.6 m. The boreholes were about the same size as the diameter of the PVC pole so that there was minimal annulus space to backfill. Menkveld (2019) used a similar approach and shows examples of how a vertical temperature profile may be contoured over time with these data to produce 2D plots. In the second approach, six CS109 (Campbell Scientific Inc.) probes were mounted on the outside of 12 mm diameter PVC rods and installed at different depths in individual boreholes. Boreholes were either hand augered or drilled with the drill rig, and then soil was backfilled around the rods, tamping periodically. This later approach was applied in the base of a topographic depression (Mannheim site) that experienced periodic ponding; the temperature sensors were installed at different depths in the vadose zone and used to assess infiltration. Figure 12 shows soil temperature fluctuations over nearly 3 annual cycles. As a third approach, the water content reflectometer sensors also included temperature monitoring capability at the one recharge plot at the Mannheim site and at the Woodlot plot at the Bethel Road Farm site (Fig. 3a; Wiebe, 2020).

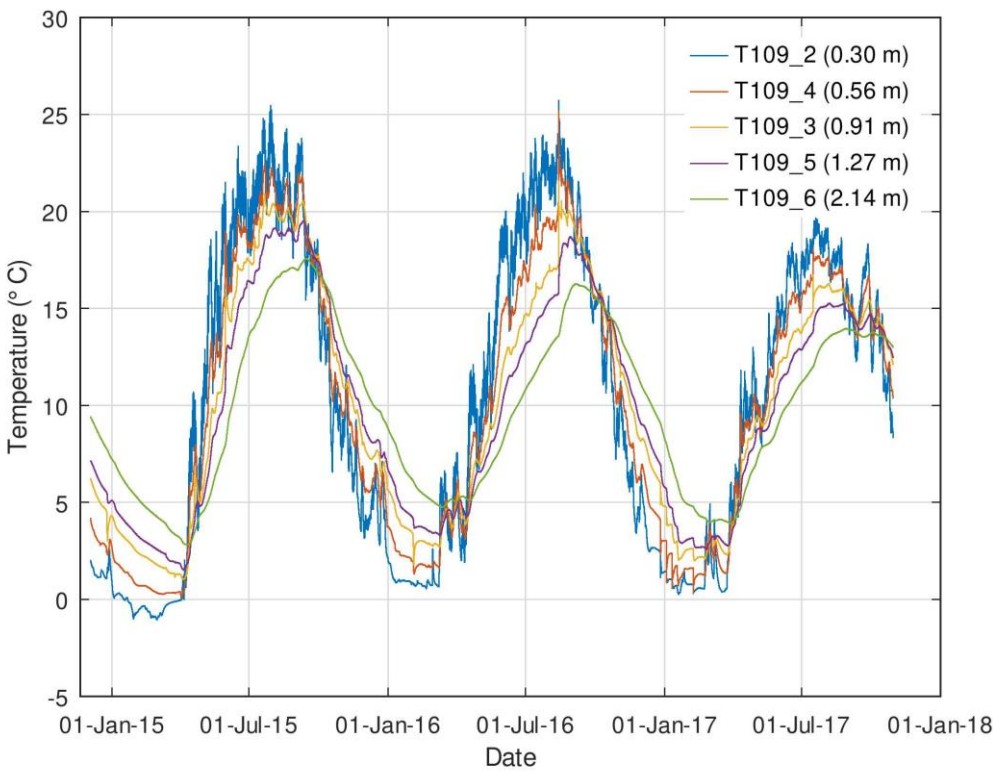

**Figure 12: Average daily soil temperatures (Wiebe et al., 2019) at five depths beneath the base of a topographic depression at the Mannheim site.**

The method of Stallman (1965) was used to estimate annual DFR (as vadose zone drainage) at the Mannheim site using data from the CS109 soil temperature sensors (Fig. 12). The idea for this analysis is mentioned by Nimmo et al. (2005) in connection with surface water but was applied here solely to soil temperatures monitored in the base of the topographic depression. The Stallman (1965) method assumes steady state flow and employs the following equations:

$$T(t,z) = T_0 + (\Delta T)\exp(-az)\sin(2\pi t/\tau - bz) \,, \tag{1}$$

$$K' = \frac{\pi C_b}{\kappa_b \tau} \,, \tag{2}$$

$$V' = \frac{q C_w}{2\kappa_b} \,, \tag{3}$$

$$a = \left[\left(K'^2 + V'^4/4\right)^{1/2} + V'^2/2\right]^{1/2} - V' \,, \tag{4}$$

$$b = \left[\left(K'^2 + V'^4/4\right)^{1/2} + V'^2/2\right]^{1/2} \,, \tag{5}$$

where $T$ is temperature as a function of time ($t$) and depth ($z$), $T_0$ is the mean temperature in surface water (i.e., soil temperature sensor at 0.3 m depth, here), $\Delta T$ is the amplitude of temperature fluctuation in surface water (i.e., soil temperature sensor at 0.3 m depth, here; either $|\max(T) - T_0|$ or $|\min(T) - T_0|$), $\tau$ is the period of fluctuation (e.g., one year), $C_b$ is the volumetric bulk heat capacity of the soil, $\kappa_b$ is the bulk aquifer thermal conductivity, $q$ is the vadose zone drainage flux, and $C_w$ is the volumetric heat capacity of water. The lower limit that the method can resolve is about $1 \times 10^{-8}$ m·s⁻¹ or about 1 mm·d⁻¹ (Stallman, 1965). The parameters $C_b$ and $\kappa_b$ were obtained from Brookfield (2009), and $C_w$ was obtained from Palmer et al. (1992). PEST (Doherty, 2015) was used to calibrate the parameters $T_0$, $\Delta T$, $q$, and a time offset factor added to $t$ to optimize the fit. Supplementary Materials Document S1 lists the parameters, input files, and GNU Octave (Eaton et al., 2019) scripts used for parameter estimation via PEST; input files for this process were created by calculating daily averages from the 15 min temperature data and then configuring the necessary input file formats required by PEST. Temperature observations from five of the six sensors (T109_2 to T109_6) were used, with equal weighting selected for simplicity. The uppermost sensor (T109_1) was not used because of its wide range of fluctuations, likely influenced by solar radiation heating the shallow soil. Figure 13 shows the results of matching the soil temperature data at three of the five sensor depths and suggests an average recharge flux of $3.5 \times 10^{-8}$ m·s⁻¹ (1,100 mm·yr⁻¹). This seems reasonable considering that local observations suggest that DFR events occur about four times per year on average (Wiebe, 2020) and that recharge rates during such events could range up to 400 mm per event (Wiebe et al., 2021).

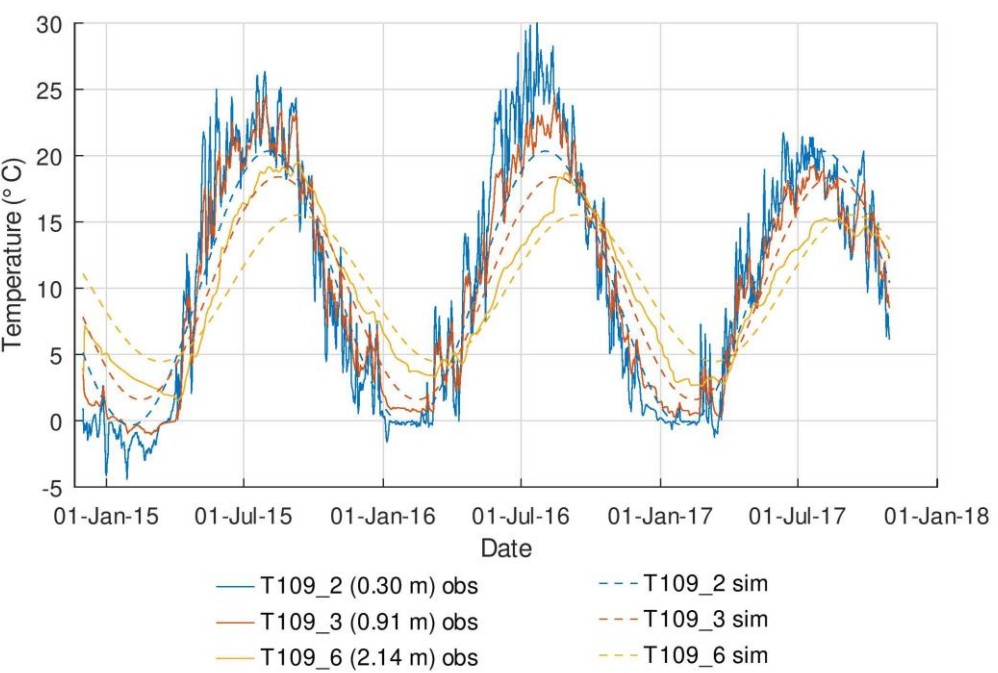

**Figure 13: Observed (obs; Wiebe et al., 2019) and simulated (sim) soil temperatures at the Mannheim site for a vadose zone drainage rate of 8.1×10⁻⁸ m·s⁻¹.**

Recharge rates can also be approximated using temperature data at shorter time scales, though the assumption of steady state requires more verification than may be necessary when using longer time scales. The Shan and Bodvarsson (2004) method was applied using the "Flux-LM" spreadsheet tool by Kurylyk et al. (2017) to estimate the vadose zone drainage rate at the Mannheim site for one day in December 2014, as an example of a time when steady state drainage conditions were approximated. The day selected (11 Dec 2014) was more than two weeks after a large infiltration event, and average daily

temperatures changed less than 0.1 °C from the previous day at each sensor. The Shan and Bodvarsson (2004) method allows estimation of the average drainage flux ($q$) through a series of soil layers. The method assumes a 1D vertical soil profile consisting of $n$ layers with values $d_i$ designating their bottom depths. The following equations related to this method are from Shan and Bodvarsson (2004). The steady state governing equation for heat transport through the system is:

$$\alpha_i \frac{d^2 T_i}{dz^2} = q \frac{dT_i}{dz} \qquad (i = 1,2,\ldots,n), \tag{6}$$

where $z$ is the depth (m), $T_i$ is the temperature (°C) at a point within layer $i$, and $q$ is the average drainage flux across all layers. The thermal diffusivity ($m^2 \cdot s^{-1}$) of the $i$th layer, $\alpha_i$, is the ratio:

$$\alpha_i = \frac{\lambda_i}{\rho c} \qquad (i = 1,2,\ldots,n), \tag{7}$$

where $\lambda_i$ is the thermal conductivity ($W \cdot m^{-1} \cdot K^{-1}$) of layer $i$, $\rho$ is the water density ($kg \cdot m^{-3}$), and $c$ is the heat capacity of water ($J \cdot m^{-3} \cdot K^{-1}$). Boundary conditions (constant temperature) for the top and bottom of the system, respectively, are:

$$T_1(d_0) = T_0, \tag{8a}$$

$$T_n(d_n) = T_B, \tag{8b}$$

where $d_0 = 0$ m and the conditions at the layer interfaces require that:

$$T_i(d_i) = T_{i+1}(d_i) \qquad (i = 1,2,\ldots,n-1). \tag{9}$$

The general solution of Eqn. 6 for temperature variations within layer $i$ is:

$$T_i(z) = C_{i.1} e^{qz/\alpha_i} + C_{i.2} \qquad (i = 1,2,\ldots,n), \tag{10}$$

where $C_{i.1}$ and $C_{i.2}$ are integral constants defined by:

$$C_{i.2} = \frac{aT_0 - T_B}{a - 1}, \tag{11a}$$

$$C_{1.1} = \frac{T_0 - T_B}{a - 1}, \text{ and} \tag{11b}$$

$$C_{(i+1).2} = e^{qd_i(1/\alpha_i - 1/\alpha_{i+1})} C_{i.1} \qquad (i = 1,2,\ldots,n-1). \tag{11c}$$

The variable $a$ is defined as:

$$a = e^{q d_n / \alpha_{eff}},$$ (12)

where $d_n$ is the overall thickness and $\alpha_{eff}$ is the effective thermal diffusivity over the $n$ layers:

$$\alpha_{eff} = d_n / \sum_{i=1}^{n} (d_i - d_{i-1})/\alpha_i.$$ (13)

Both a one− (thermal) layer model and a two− (thermal) layer model were tested, with slightly better results in terms
of root mean square error (0.09 °C vs 0.13 °C) from the two−layer model. Data (Table 6) from the six CS109 sensors were
applied with thermal conductivity layer estimates of 1.0 W·m$^{-1}$·°C$^{-1}$ for the silty uppermost 0.8 m layer of soil, and a value of
2.0 W·m$^{-1}$·°C$^{-1}$ was applied to the underlying gravelly and sandy materials. These thermal conductivity values were chosen to
be generally consistent with the literature for different soil types (e.g., Stonestrom and Constantz, 2003). The flux magnitude
was estimated to be $1.2 \times 10^{-7}$ m·s$^{-1}$ (10 mm·d$^{-1}$), and the observed and simulated results are shown in Fig. 14.


**Table 6: Temperature data (Wiebe et al., 2019) used for Flux-LM (Kurylyk et al., 2017) vadose zone drainage estimate for 12:00 pm on 11 Dec 2014.**

| Site | Sensor name | Depth [*] (m) | Temperature (°C) |
|---|---|---|---|
| Mannheim | CS109-1 | 0.08 | -0.966 |
| | CS109-2 | 0.30 | 1.219 |
| | CS109-4 [†] | 0.56 | 3.154 |
| | CS109-3 [†] | 0.91 | 5.355 |
| | CS109-5 | 1.27 | 6.425 |
| | CS109-6 | 2.14 | 8.760 |

[*] Depth of bottom of sensor (0.06 m length)

[†] Please note that the numbering order of the sensors differs slightly from the depth order of the sensors.

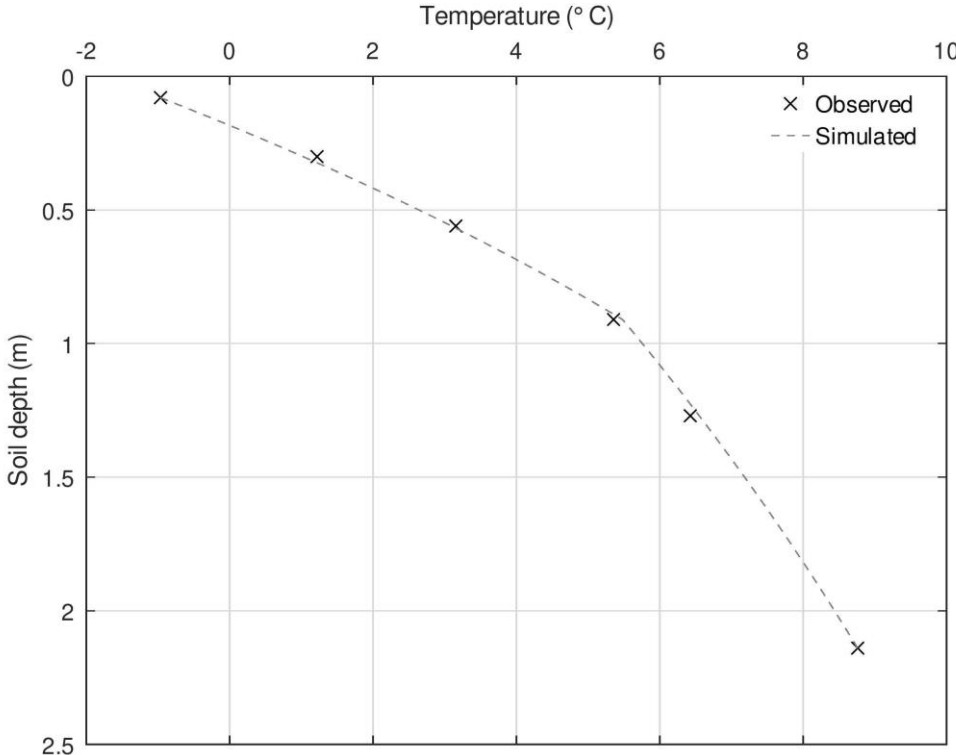

**Figure 14: Observed (Wiebe et al., 2019) and simulated soil temperatures at the Mannheim site for 12:00 pm on 11 Dec 2014 for a**
**vadose zone drainage rate of $1.2\times10^{-7}$ m·s$^{-1}$.**

## 6 Creek data

Rating curves were compiled from manual stream gauging measurements with a wading rod instrument. The curves mostly captured low and moderate flow conditions and are generally lacking high flow conditions. One high flow condition was
roughly estimated for the Huron Road Farm site during an April snowmelt event. Figure 15 shows the rating curves at five sites along the creek. Creek water levels and temperatures were recorded at several of these sites. For example, Fig. 16 shows creek water levels at the Huron Road Farm site and streamflow estimated via the rating curve for the site. Either vented (PT12®; AquiStar Inc.) or non−vented (Levelogger®; Solinst Inc.) pressure transducers were used at the stream stations.

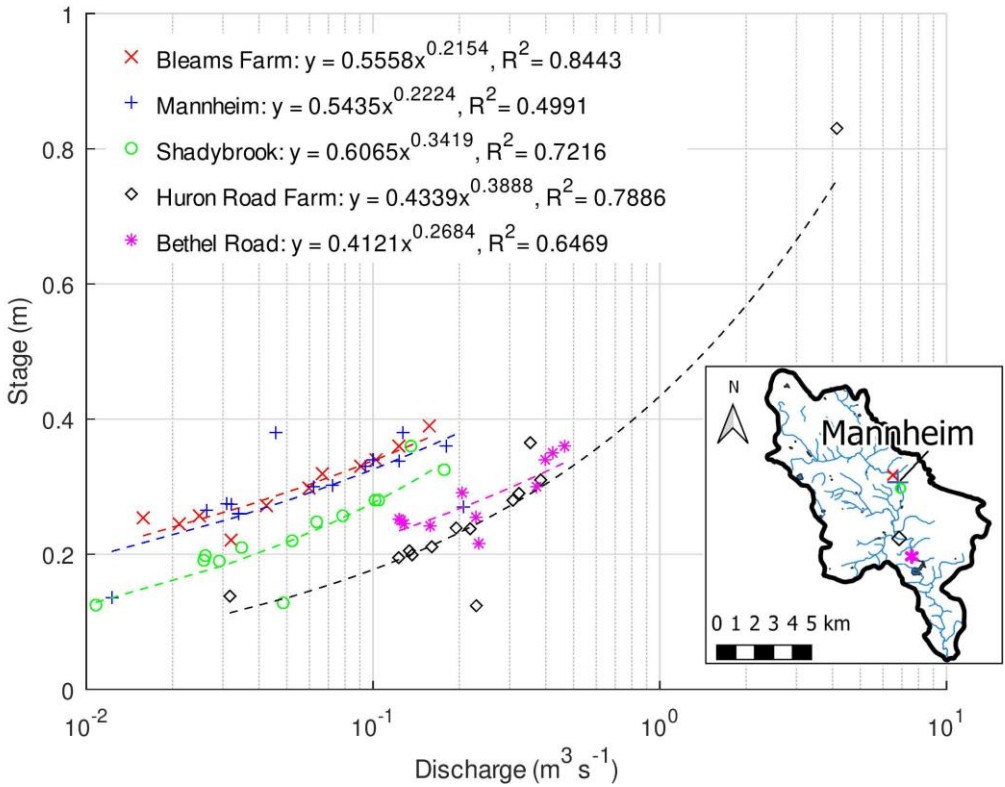


**Figure 15: Rating curves (Wiebe et al., 2019) at five locations along the creek (map: DMTI, 2011; Grand River Conservation Authority, 1998). The curves were developed from occasional manual measurements of streamflow; water levels were recorded electronically on a more consistent basis.**

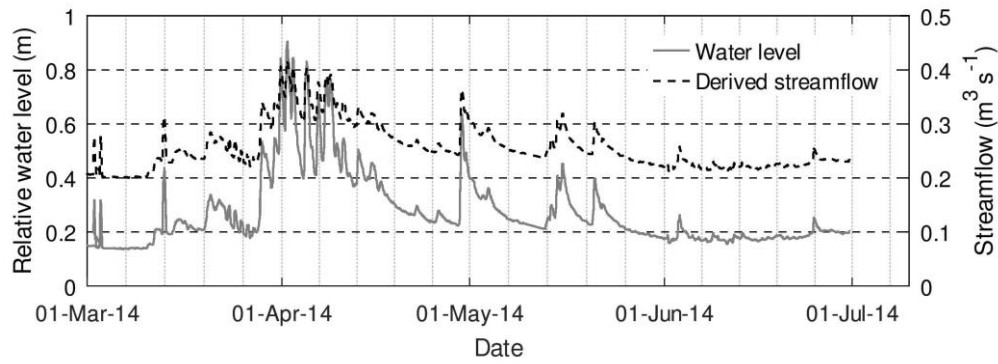


**Figure 16: Creek water levels (Wiebe et al., 2019; subsampled at a 1-hour time scale) and derived streamflow estimates.**

## 7 Geochemistry data

Sampling was conducted to record snapshots of the geochemistry of Alder Creek, snowpack within the watershed, and groundwater. Samples were analyzed for major cation and anion concentrations and for O−18 and H−2 isotopes. Nitrate, chloride, sulfate, dissolved oxygen, pH, and turbidity data were collected at several locations within the creek during the summer of 2013. Snow and creek samples were collected and analyzed for O−18 and H−2 isotopes as well as nitrate, chloride, sulfate, soluble reactive phosphorus (SRP), and total phosphorus concentrations during Feb−Apr 2014. Figure 17 shows total phosphorus and SRP concentrations in the creek at five sites from Mar−Jun 2014. Other studies (Ontario Ministry of the Environment, 2012; Irvine, 2018) in southern Ontario have suggested similar general patterns, though the sparsity of the sampling times here somewhat hinders comparison. Fig. 18 shows isotope data for creek, groundwater, and snow samples. The creek and groundwater isotopes align closely, reflecting the role of groundwater discharge in maintaining baseflow in winter. The groundwater isotopes are more enriched in the heavier isotopes than the snowpack samples, illustrating the greater contribution of rainfall to groundwater recharge.

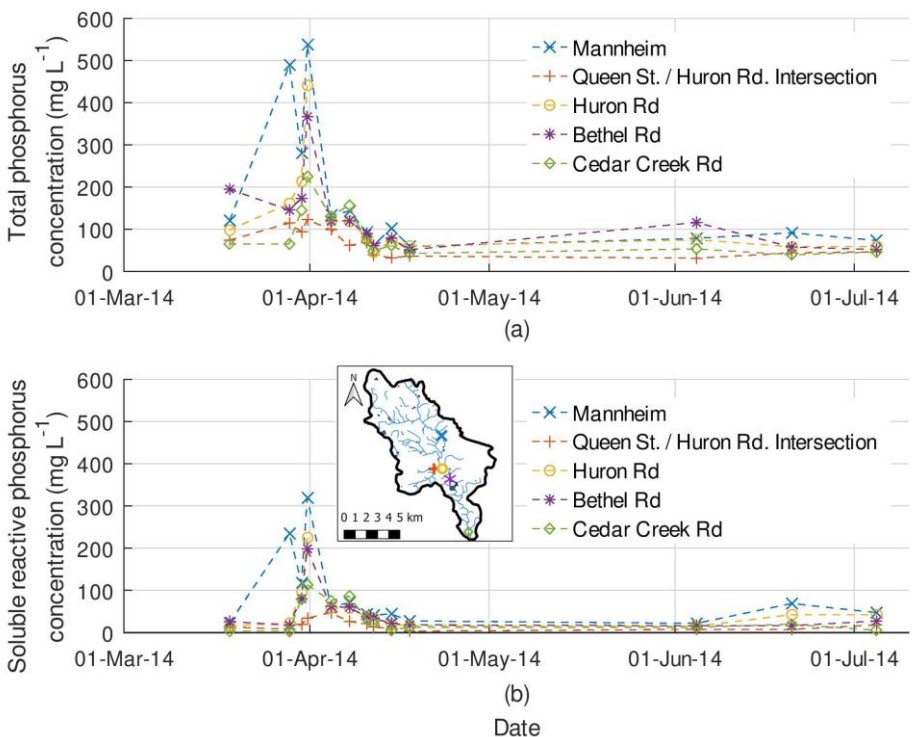

**Figure 17: (a) Total phosphorus and (b) SRP concentrations at five sites along the creek in Mar−Jun 2014 (Wiebe et al., 2019; map: DMTI, 2011; Grand River Conservation Authority, 1998). Fig. 16 shows the creek water levels and flow estimates during this time period.**

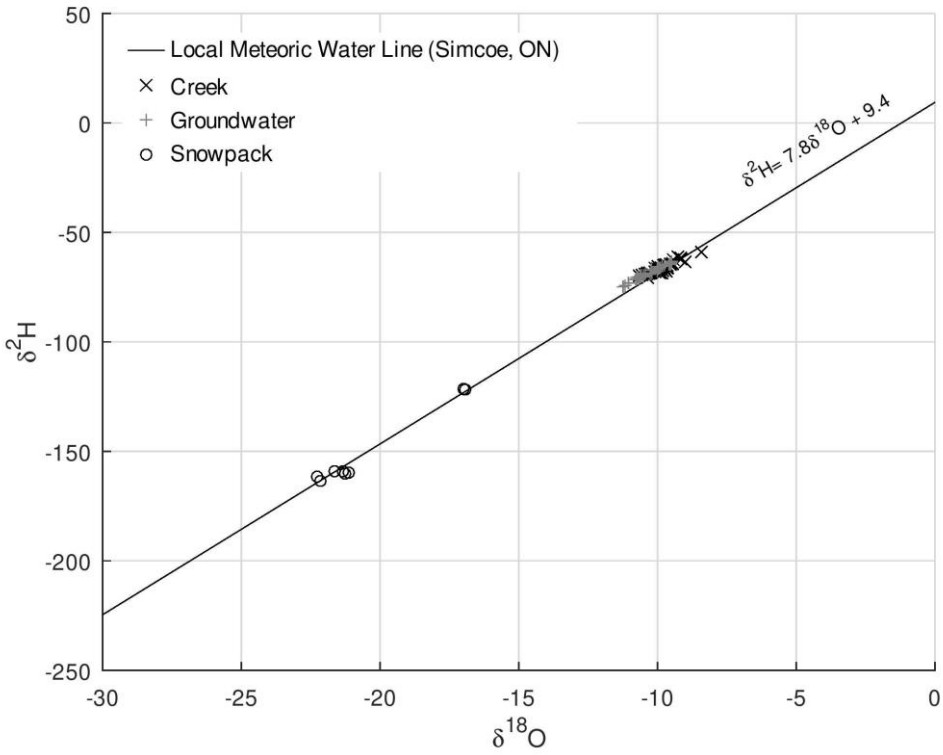

**Figure 18: Isotope data from the watershed (Wiebe et al., 2019; local meteoric water line from Bajc et al., 2018). Groundwater and creek results coincide and show a contrast from snowpack results, suggesting that recharge is mostly derived from rainfall.**

## 8 Data availability

Apart from the data shown in Fig. 2, all data presented in this paper are available or may be derived from the Federated Research Data Repository (https://doi.org/10.20383/101.0178; Wiebe et al., 2019). Equipment or sampling locations corresponding to where the data were collected are provided in several GIS shapefiles included in the dataset. The files of the dataset may be downloaded without creating a user account by right–clicking the individual files of interest, selecting "Save Link As...", and preserving the file extensions. Table 7 summarizes the time periods associated with the data.

Several papers and theses have employed the data presented above: Hillier (2014), Menkveld (2019), Wiebe and Rudolph (2020), Wiebe (2020), and Wiebe et al. (2021).

Additional datasets for the area are available from the sources listed in Table 8.

**Table 7: Data availability for field stations of the Southern Ontario Water Consortium – Alder Creek project.**

| Parameter(s) | Site/Location | Time Period | Time step |
|---|---|---|---|
| Air temperature and relative humidity | WS1 – WS7 | May 2013 − Dec 2018 | 15 min |
| Wind speed and direction | WS1 – WS7 | May 2013 − Dec 2018 | 15 min |
| Rainfall | WS2 – WS7 | May 2013 − Dec 2018 | 15 min |
| Snowfall | WS2 – WS7 | Nov 2014 − Apr 2015 | 15 min |
| Solar radiation | WS1 – WS7 | May 2013 − Dec 2018 | 15 min |
| Barometric pressure | WS4<br>WS7 | May 2015 − Apr 2018<br>Dec 2013 − Apr 2017 | 15 min |
| Manual water levels | all wells | Jan 2014 − Dec 2018 | Occasional |
| Pressure transducer water levels and temperatures (observation wells) | Mannheim, Bethel Farm<br>Huron Rd. Farm (CMT4) | Nov 2014 − Apr 2018<br>Mar 2014 − Dec 2018 | 15 or 30 min |
| Soil moisture and electrical conductivity | Mannheim – TDR system<br>Mannheim – CS655 system [†]<br>Bethel Rd Farm – CS655 [†] | Nov 2014 − Jun 2018<br>Dec 2015 − Jun 2018<br>Jun 2016 − Jul 2019 | 15 min |
| Soil temperature | Mannheim – TidbiT poles<br>Mannheim – CS109 | Nov 2017 − Apr 2018<br>Nov 2014 − Jun 2018 | 15 min |
| Relative barometric pressure [*] | Mannheim<br>Bethel Rd Farm | Nov 2015 − Oct 2017<br>Mar 2016 − May 2018 | 15 min |
| Creek water levels and temperatures | Mannheim – WL5 North<br>Mannheim – transect PT12<br>Mannheim − RR1<br>Huron Rd. Farm<br>Bethel Rd Bridge<br>Cedar Creek Rd Bridge | Jul 2014 − Jun 2017<br>Nov 2014 − Apr 2018<br>Nov 2013 − Apr 2018<br>Aug 2013 − May 2014<br>Aug 2013 − Jun 2015<br>Sep 2014 − Dec 2016 | 5 or 15 min |
| Anion concentrations ($Cl^-$, $SO_4^{2-}$, $NO_3^-$) | Consistently at 5 locations along creek | 4× from Jul − Aug 2013, and 14× from Mar−Jul 2014 | Occasional |
| Total phosphorus and soluble reactive phosphorus | 5 locations along creek | 14× from Mar−Jul 2014 | Occasional |
| $\delta^{18}O$ and $\delta^2H$ isotope concentrations in creek, groundwater, and snow | Multiple locations along creek; Mannheim CMT wells; snow at WS1 to WS6 | 10× from Jul 2013 − Feb 2014 | Occasional |
| Temperature, electrical conductivity, total suspended solids, total dissolved solids, specific conductivity, salinity, non−linear function electrical conductivity, dissolved oxygen, barometric pH, turbidity | Consistently at 5 locations along creek | 14× from Mar−Jul 2014 | Occasional |
| Anion ($Cl^-$, $NO_2^-$, $NO_3^-$, $PO_4^{3-}$, $SO_4^{2-}$) and cation ($Na^+$, $NH_4^+$, $K^+$, $Mg^{2+}$, $Ca^{2+}$) concentrations | Snow at WS1, WS3, WS4, WS6; creek samples from various locations | Feb 2014 (snow and creek), Mar 2014 (creek) | Occasional |

[*] For correcting non−vented pressure transducers; Solinst Barologger data for the Mannheim and Bethel Road Farm sites may also be used for corrections

[†] Also includes temperature

**Table 8: Publicly available data from other sources that are complementary to the Wiebe et al. (2019) dataset.**

| Type of data | Reference and Web URL | File Type |
|---|---|---|
| Surficial geology and stratigraphic subsurface layers | Bajc and Shirota (2007) <br> *http://www.geologyontario.mndm.gov.on.ca/index.html* | Google Earth™ |
| Ground surface elevation* | Ontario Ministry of Northern Development, Mines, Natural Resources and Forestry (2019) <br> *https://geohub.lio.gov.on.ca/maps/mnrf::ontario-digital-terrain-model-lidar-derived/about* | IMG Raster |
| Land use† | Grand River Conservation Authority (2017a) <br> *https://data.grandriver.ca/downloads-geospatial.html* | TIF Raster |
| Streamflow | Water Survey of Canada (2019) <br> *https://wateroffice.ec.gc.ca/mainmenu/historical_data_index_e.html* | CSV |
| Weather data | Government of Canada (2019) <br> *http://climate.weather.gc.ca/historical_data/search_historic_data_e.html* | CSV |
| Weather data | Seglenieks (2020) <br> *http://weather.uwaterloo.ca/data.html* | CSV |
| Watersheds within the Grand River basin† | Grand River Conservation Authority (2017b) <br> *https://data.grandriver.ca/downloads-geospatial.html* | GIS Shapefile |
| Water courses† and water bodies† | Grand River Conservation Authority (2022a,b) <br> *https://data.grandriver.ca/downloads-geospatial.html* | GIS Shapefile |
| Water budget and risk assessment modelling | Matrix and S.S. Papdopulos Associates Inc. (2014b) <br> *https://www.sourcewater.ca/en/source-protection-areas/region-of-waterloo-tier-3.aspx* | Report |

* Contains information made available under Open Government Licence – Ontario, v1.0 (https://www.ontario.ca/page/open-government-licence-ontario).

† Contains information made available under Grand River Conservation Authority's Open Data Licence v2.0 (https://data.grandriver.ca/about-licensing.html).

**9 Code availability**

Supplementary Materials Document S1 contains background information for the optimization of the parameters for the Stallman (1965) method. The file includes GNU Octave (Eaton et al., 2019) scripts and file formats used to conduct parameter estimation via PEST (Doherty, 2015).

**10 Summary**

Hydrological and meteorological instruments were deployed in the Alder Creek watershed between 2013 and 2018. This watershed provides source water to several well fields, and the data have been used within numerical models estimating

groundwater recharge. A new analysis of vertical soil temperature profile records presented above suggested that annual drainage rates related to ponding in the base of a topographic depression at the Mannheim site could be around 1,100 mm per year from 2015 to 2017. Despite the short duration of the data collection (3 to 4 years), it is hoped that the data may be useful to other researchers and instructors.

**Supplement link**

Supplementary Materials Document S1.pdf ("GNU Octave code and PEST file formats employed for vadose zone drainage calculations").

**Author Contribution**

Conceptualization, AW; methodology, AW; software, AW; formal analysis, AW; investigation, AW; resources, DR; data curation, AW; writing – original draft preparation, AW; writing – review and editing, DR, AW; visualization, AW; supervision, DR; project administration, DR; funding acquisition, DR, AW.

**Competing interests**

The authors declare that they have no conflict of interest.

**Acknowledgements**

We acknowledge the support of the Natural Science and Engineering Research Council of Canada (NSERC; IPS Grant #485430 to A.J. Wiebe, and Discovery Grant to D.L. Rudolph). The Global Water Futures project of the Canada First Research Excellence Fund provided the resources for publishing the dataset online. Field equipment was provided by the Southern Ontario Water Consortium (Ontario MEDI, Project #21616; FedDev Ontario, Project #801680). Thanks to the many students and technicians who assisted with the field work and data collection: P. Johnson, B. Ingleton, O. Idika, S. McKay, J. Dickhout, K. Blowes, L. Harrison, C. Hillier, J. Robertson, J. Elliott, E. Mesec, P. Menkveld, N. Couperus, N. Long, J. Stevens, I. Mercer, E. Pai, E. Huang, S. Indris, J. Ju, J. Leon, and A. Wicke. Thanks to E. Mesec, E. Pai, P. Menkveld, and E. Huang for assisting with borehole logging, to E. Huang for conducting the grain size analyses, and to E. Mesec and L. Harrison for collecting the geochemistry and isotope samples. We are indebted to the individuals, businesses, and local governments who agreed to host field equipment during the study: R. and W. Goettling, G. and L. Kaster, D. and P. Mighton, the Region of Waterloo, Colour Paradise Greenhouses, Herrle's Farm Market, Nith Valley Organics, Rebel Creek Golf Club, and the County of Oxford.

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
