# Peer review of "Meteorological and hydrological data from the Alder Creek watershed, SW Ontario"

_Earth System Science Data, 2022_

## Author Response (AR1)

**Referee Comments and Authors' Responses – Open Discussion**

The authors would like to thank Anonymous Referee #1 for the comments and questions. In the following, the reviewer's comments are in regular text, and the authors' responses are in Italics and follow the word "Response." All responses are the same as in the posted response to the comments (AC1: 'Reply to RC1'), except in the case of specific comment # 10, which has been revised. Changes to the manuscript are indicated following the heading "Changes" when these were made. All line and figure numbers refer to the original, preprint manuscript.

RC1: 'Comment on essd-2022-46', Anonymous Referee #1, 19 Mar 2022

Citation for Referee Comments: https://doi.org/10.5194/essd-2022-46-RC1

Citation for Original Response from the Authors : https://doi.org/10.5194/essd-2022-46-AC1

Review for ESSD-2022-46

Title: Meteorological and hydrological data from the Alder Creek watershed, SW Ontario

This study provides meteorological and hydrological datasets in the Alder Creek watershed by monitoring precipitation, air temperature, solar radiation, groundwater levels, soil texture, soil moisture, soil temperature, streamflow, and geochemistry. This comprehensive dataset can be useful for systematically studying hydrological processes in the Alder Creek watershed. However, there still exist some issues in this stage.

1. My major concern for this study is how typical or special is the Alder Creek watershed, and is this region interesting enough to attract audiences of ESSD to use this dataset. What kinds of unique studies can be conducted in this small watershed rather than any other watersheds?

Response: The Alder Creek watershed represents many small watersheds where there are competing pressures related to groundwater. The watershed has multiple types of land use, including agriculture, aggregate/sand and gravel extraction, and urban areas. These land use types each have their own groundwater quality concerns. Expanding urban development within the watershed is a major concern and a potential influence on groundwater recharge rates. Multiple public well fields are located within the watershed or capture water recharged within it, and these rely on maintaining groundwater recharge quantity and quality. There are ecological concerns regarding groundwater baseflow to the creek and how the public wells may influence this. Surficial geology data, stratigraphic data, and land use data are available for the watershed. Thus, the watershed is useful for assessing various critical issues related to groundwater management due to the many important issues related to the watershed, and the amount of data available. *Changes: The Introduction and Site Description sections have been updated to include this information, and the available datasets from other organizations are now listed in a new table in Section 8.*

2. This manuscript is more like a report to list all the hydrological data one by one. The measurements and characteristics of each hydrological data were described in detail, but these hydrological data were not connected together to provide any new knowledge and understanding. There are no clear result and discussion parts in this manuscript. Analysis of these data is lacked. Based on this dataset, can you provide any interesting characteristics of the hydrological processes, such as the interaction between climate and groundwater, at the site level or watershed level?

Response: The goal of this data article was to present the available hydrological data and provide examples of how the data may be used. This format has been used by other researchers in their data articles regarding Canadian watersheds (e.g., Rasouli et al., 2019, ESSD, https://doi.org/10.5194/essd-11-89-2019; Fang et al., 2019, ESSD, https://doi.org/10.5194/essd-11-455-2019; Gibson et al., 2020, Data in Brief, https://doi.org/10.1016/j.dib.2020.105308 2352-3409; and Spence and Hedstrom, 2018, ESSD, https://doi.org/10.5194/essd-10-1753-2018). The work of connecting the data and generating new understanding is briefly included in the manuscript (new temperature modelling work to estimate recharge rates) but is more prominent in the theses and publications that have employed the data. The intention here is that future studies related to similar watershed environments will be better able to use these data as a result of the descriptions in this manuscript. Results and Discussion sections were not included, in keeping with the format precedence interpreted from the other studies. References to studies that have used various aspects of the presented data for specific scientific research have been listed in the document (Lines 380-381) and some have been referred to directly. For example, the data have been used (Wiebe and Rudolph, 2021; Wiebe, 2020) to study the spatial correlation of rainfall across the watershed (mentioned on Line 132). The spatial correlation of rainfall at this scale could not be evaluated without the data from the multiple weather stations installed in the watershed.

*The interaction between climate and groundwater could be modelled based on the data. Wiebe (2020) provides an example of a Monte Carlo analysis. Other analyses would be possible.*

**Specific issues:**

**Introduction**

1. Line 18-19: "Comprehensive meteorological and hydrological data from multiple field stations within small to mid-sized watersheds are seldom publicly available." The USGS provides hydrological data across the country and there may exist many sites located in small to mid-sized watershed.

Response: The unique part of the Alder Creek dataset is that there are data from multiple weather stations installed in and around the small to mid-sized watershed. There are indeed some watersheds with multiple weather stations within a small area that have been reported in the scientific literature (e.g., Walnut Creek, Iowa – Chaplot et al., 2005, J. Hydrol., https://doi.org/10.1016/j.jhydrol.2005.02.019), but this does not appear to be common. Characteristics of different watersheds can be highly variable and it is anticipated that there is value in having these types of dense meteorological data provided for multiple watershed settings.

2. Line 30. There is an extra "a".

*Response/Changes: This has been revised, thanks.*

3. Line 33. The Alder Creek watershed is important for local supply. Aside from this local importance, are there any other characteristics that make this small watershed be an interesting place to conduct hydrological studies that have broad influence in science.

*Response: Please see the response to Comment (1.) in the General Comments section at the top.*

**Site description**

4. Are all the datasets first monitored and published in this study? The Water Survey of Canada is mentioned, so is the stream data collected from this source. If so, a table that lists all the data features and sources (collect, simulate, or monitor) may be helpful for audience to have an overview of these datasets.

Response: This is a good idea. Most of the data discussed were first published in the cited Wiebe et al. (2019) dataset. Streamflow data (Water Survey of Canada), temperature and precipitation data (Environment Canada), and other weather data (University of Waterloo weather station) are mentioned in the text.

Changes: These complementary sources of information have now been listed in a new table in section 8, along with the datasets referenced in the material added in response to Comment (1.) in the General Comments section at the top.

**Groundwater data**

5. Figure 9. The label of this figure is missing.

Not sure what is missing here. Would you be able to clarify? The figure is referenced in the text on Line 167, the caption appears to be present in Lines 177-180, and the two axes are labelled.

**Vadose zone data**

6. Does soil texture in different sites have any impacts on soil temperature and moisture?

Response: Soil texture and surrounding vegetation have an impact on soil moisture and soil temperature at the Mannheim and Bethel Road Farm sites. Soil moisture readings at the Mannheim site (for both the TDR and CS655 sensors) are higher than the readings at the Bethel Road Farm site. This aligns with soil texture differences between the fine-grained upper soil layers at the Mannheim site, and the sandy soil in which the sensors are installed at the Bethel Road Farm site. The maximum and minimum soil temperatures are generally similar at both sites.

**Soil moisture**

7. Why there is no figure to show the soil moisture data.

Response: Good catch.

Changes: An example of these data has been added.

**Soil temperature**

8. Figure 14. The simulated results seem underestimate peak values, especially in T109\_2 and T109\_3, what is the potential reason

Response: The results minimized the combined sum of squared errors from all five temperature observation depths (only three curves are shown for the sake of clarity). There is somewhat of a trade-off between increasing the flux to match the peaks of the shallower sensors, and decreasing the flux to match the deepest sensor. This may indicate a slight mismatch of the assumptions of 1D flux with the actual field conditions.

**Creek data**

9. Is discharge monitored or just estimated according to stream water level?

*Response: Only stream water level was monitored on a regular basis. Rating curves developed from temporal spot measurements (stream gauging) were used to estimate the streamflow.*

**Geochemistry data**

10. Figure 18. There are few samples during May and Jun. The variations of P concentrations can be significant in a short time period according to the data around April.

Response: We previously responded (https://doi.org/10.5194/essd-2022-46-AC1) with speculation that the peaks in the total P and SRP in March and April likely coincide with the melting of the snowpack and the process of overland flow over frozen soils, suggesting that there might be less variation in the total P and SRP runoff after the ground thaws. While other studies in southern Ontario have suggested patterns such as the one alluded to in this figure, the exact mechanisms may not be clear or may differ in different settings. Therefore, we have amended our response and agree that the timing of the samples leads to uncertainty.

*Changes: A comment noting the similarity of the general pattern to two studies in the literature was added to the text, with mention of the sparsity of the data points.*

**11. L360-365. What conclusions can be made according to the isotopes data?**

*Response: Thanks for the suggestion Changes: The following has been added to the manuscript:*

> The creek and groundwater isotopes align closely, reflecting the role of groundwater discharge in maintaining baseflow in winter. The groundwater isotopes are more enriched in the heavier isotopes than the snowpack samples, illustrating the greater contribution of rainfall to groundwater recharge.

The authors would like to thank Anonymous Referee #2 for the comments and questions. In the following, the reviewer's comments are in regular text, and the authors' responses are in Italics and follow the word "Response." All responses are the same as in the posted response to the comments (AC2: 'Reply to RC2'). Changes to the manuscript are indicated following the heading "Changes" when these were made. All line, table, and figure numbers refer to the original, preprint manuscript.

**Anonymous Reviewer #2 (RC2)**

Citation for Referee Comments: https://doi.org/10.5194/essd-2022-46-RC2

Citation for Authors' Responses: https://doi.org/10.5194/essd-2022-46-AC2

In this manuscript, the authors have provided a report on hydrometeorological data from a watershed in southwestern Ontario, Canada. The dataset provided is comprehensive, which is very useful for different environmental disciplines. The paper is well-written. But, it needs some revision before it can be considered for publication. Below are some comments that can be used to improve the manuscript.

There are so many figures, maybe some of them can be deleted. For instance, figures 6, 8, 9, 15, 16 can be safely removed. Merge Figures 3 and 4. Merge Figures 10 and 11. Why do we need simulations included in this manuscript (i.e., Figure 14)? I recommend removing all simulations from the manuscript and just focusing on the observations that are not available from national monitoring networks.

**Response - Figures:**

Figures are often an early point of interaction for the reader to assess the gist of a manuscript (or dataset, in this case). We propose to maintain most of the figures and remove Figure 9 for the reasons stated below. We will reduce and optimize the use of the figures within the manuscript.

*Figure 6 shows one of the highlights of the dataset – rainfall measurements at multiple stations within a small watershed. We propose to keep this figure.*

*Figure 8 – This figure illustrates that the water level data have not been corrected (which might be erroneously assumed). We propose to keep this figure.*

The legibility of Figures 10 and 11 will likely decrease if they are merged and each borehole's size in the image becomes smaller, so we would recommend against this.

Figure 15 – While this figure does not present a large amount of information, if the simulations remain in the manuscript (as argued below), then it is a helpful illustration. We propose to keep this figure.

*Figure 16 is helpful for showing the extent of the rating curve data (i.e., mostly at low flows). We propose to keep this figure.*

**Response - Simulations:**

The simulations illustrate types of analyses that could be performed with the dataset. One of the goals of ESSD is to provide interesting and useful articles (Carlson and Oda, 2018, ESSD, https://doi.org/10.5194/essd-10-2275-2018). We would prefer to keep these in the manuscript for those interested in groundwater recharge, and to promote the under-utilized (Kurylyk and Irvine, 2019, Ground Water, https://doi.org/10.1111/gwat.12910) use of temperature data for these types of estimates.

Changes: Figures 3 and 4 have been merged.

Figure 9 has been removed as suggested.

In general, all captions are too short (e.g., Table 7). Provide more descriptive captions for all figures and tables. Add another column for the time steps of the parameters.

*Response/Changes: Many of the figure and table captions have been expanded with additional information.*

L33: Spell out acronyms i.e., CH2MHILL, SSPA. The same goes with TRCA on line 41, OMNRF on line 51, OGS on line 53, OMNR on line 54. And also, why does OMNRF change to OMNR?

*Response: OK. OMNR to OMNRF: The Ontario Ministry of Natural Resources changed its name to include Forestry.*

Changes: CH2MHILL is a company name, but the others have now been spelled out.

L43: It was mentioned "between 2013 and 2018" in the abstract. Why is it different here?

*Response/Changes: This has been fixed; thanks for the careful review.*

Fig. 2: Is there any data on groundwater recharge? Without infiltration, we cannot have the water budget closed.

*Response/Changes: The average recharge rate for the watershed has now been mentioned here, citing previous work.*

L82: It shouldn't be "were" instead of "was"?

Response: Correct, thanks.

Changes: This has been changed.

Fig.3: What does "Temperature sensor" refer to? Soil or air? Mention explicitly.

*Response/Changes: The legend item refers to a soil temperature sensor – this has been clarified.*

Table 1: Spell out "SOWC" in the caption.

Response: OK

Changes: This has been done.

L166: What does "Temperature" refer to? Soil, air, or water?

Response/Changes: Line 166 refers to groundwater temperature – this has been clarified.

---

## Author Response (AR2)

*The authors are grateful for the thoughtful review by Anonymous referee # 1 (19 May 2022). In the following, Anonymous referee #1's comments are in regular font, and the authors' responses to the comments are in blue Italics following the heading, "Authors' Responses". Changes to the manuscript are then described under the heading, "Changes to the manuscript".*

Report #1
Submitted on 19 May 2022
Anonymous referee #1

The authors have addressed most of the comments. The manuscript has been improved after revision. I just have another two comments:

The authors indicated that one interesting study that could be conducted in the Alder Creek watershed is the impact of land-use change on groundwater quality/quantity. What are the corresponding land use types of observation wells that provide groundwater data? Is this data could be provided.

*Authors' Response:*

*The publicly available Grand River Conservation Authority (2017a) dataset (mentioned in Table 8) shows land use as a raster map with 20 m by 20 m cells. Older land use maps or possibly air photos may be available for comparison. For example, the Southern Ontario Land Resource Information System, (*https://data.ontario.ca/dataset/southern-ontario-land-resource-information-system*) compiled by the Ontario Ministry of Natural Resources shows a snapshot of land use around 2000 (vector GIS data). The locations of the observation wells in the GIS data portion of the Wiebe et al. (2019) dataset could be overlaid on this map to estimate the land use at each location. Some work on the topic of land use change has been performed by Matrix and S.S. Papdopulos Associates Inc. (2014b), one of the references mentioned in Table 8.*

*Table 8 lists the most current information for the watershed and would be a starting place for an in-depth study. Other sources of information than the ones listed in Table 8 would include data that are not immediately publicly available (i.e., not posted on a website) but would be available by request (for instance, from the Regional Municipality of Waterloo). MSc and PhD theses from the University of Waterloo (available online at:* https://uwspace.uwaterloo.ca/*) may also contain additional information for the Alder Creek watershed.*

*We believe that overlaying the GIS locations of the observation wells on the land use dataset listed in Table 8 is relatively straight-forward and would answer the question posed by Anonymous referee #1. For this reason, we recommend no further changes to the manuscript.*

*Changes to the manuscript:*

*None*

The melting of the snowpack is important for hydrological processes in the Alder Creek Watershed. Could more information be given regarding the melting of snow (e.g. timing)?

*Authors' Response:*

*Thanks for this suggestion. The timing of snowmelt (on a daily basis) has previously been estimated for the Alder Creek watershed by Wiebe et al. (2021) based on the degree-day method discussed by Rango and Martinec (1995).*

*Changes to the manuscript:*

*A note referring to the Wiebe et al. study has been added at line 168 within the paragraph that discusses snowfall measurements. The two new sentences refer to a hydrology textbook (Dingman, 2015) and to a paper by Rango and Martinec (1995) on calculating snowmelt. Both new references have been added to the list of references.*